# Directional Sheaf Hypergraph Networks: Unifying Learning on Directed and Undirected Hypergraphs

**Emanuele Mule**[1][*]   **Stefano Fiorini**[2]   **Antonio Purificato**[1,3][†]   **Federico Siciliano**[1]
**Stefano Coniglio**[4]   **Fabrizio Silvestri**[1]

[1] Sapienza University of Rome, Rome, Italy
[2] Independent Researcher
[3] Amazon Research
[4] University of Bergamo, Bergamo, Italy

## Abstract

Hypergraphs provide a natural way to represent higher-order interactions among multiple entities. While undirected hypergraphs have been extensively studied, the case of directed hypergraphs, which can model oriented group interactions, remains largely under-explored despite its relevance for many applications. Recent approaches in this direction often exhibit an implicit bias toward homophily, which limits their effectiveness in heterophilic settings. Rooted in the algebraic topology notion of Cellular Sheaves, Sheaf Neural Networks (SNNs) were introduced as an effective solution to circumvent such a drawback. While a generalization to hypergraphs is known, it is only suitable for undirected hypergraphs, failing to tackle the directed case. In this work, we introduce *Directional Sheaf Hypergraph Networks* (*DSHN*), a framework integrating sheaf theory with a principled treatment of asymmetric relations within a hypergraph. From it, we construct the *Directed Sheaf Hypergraph Laplacian*, a complex-valued operator by which we unify and generalize many existing Laplacian matrices proposed in the graph- and hypergraph-learning literature. Across 7 real-world datasets and against 13 baselines, DSHN achieves relative accuracy gains from 2% up to 20%, showing how a principled treatment of directionality in hypergraphs, combined with the expressive power of sheaves, can substantially improve performance.

## 1 Introduction

Learning from structured, non-Euclidean data has been dominated by Graph Neural Networks (GNNs), which propagate and aggregate features along pairwise edges (Scarselli et al., 2009; Asif et al., 2021). Sheaf Neural Networks (SNNs) (Hansen & Gebhart, 2020; Bodnar et al., 2022) extend GNNs by leveraging the algebraic concept of a *cellular sheaf*. They assign vector spaces to nodes and edges, along with learnable *restriction maps* propagating information between them. By operating in higher-dimensional feature spaces, SNNs mitigate oversmoothing and enhance performance on heterophilic graphs, where neighboring nodes exhibit dissimilar features (Purificato et al., 2025).

While effective on graphs, both GNNs and SNNs are inherently limited to dyadic relations. Many real-world systems such as social networks (Benson et al., 2016; 2018a), biological systems (Traversa et al., 2023), and protein interactions (Murgas et al., 2022) exhibit multi-way relationships that cannot be captured by pairwise links alone. Hypergraph Neural Networks (HGNNs) address this limitation by modeling hyperedges as sets of nodes, enabling the learning of multi-entity dependencies (Murgas et al., 2022; Chen et al., 2022). However, traditional HGNNs face two main limitations. First, they inherit fundamental drawbacks from their graph-based counterparts: many architectures assume homophily, which is often violated in heterophilic settings, and are prone to

---

[*]Corresponding author: `emi.mule2001@gmail.com`.
[†]Work done outside of the company.

oversmoothing, where deep message passing causes node representations to converge and lose discriminative power (Li et al., 2025; Telyatnikov et al., 2025; Chen et al., 2022; Nguyen et al., 2023). Second, most HGNNs are formulated for undirected hypergraphs, treating hyperedges symmetrically and neglecting orientation even when such hyperedges encode asymmetric or causal relationships such as chemical reactions, metabolic pathways, and causal multi-agent interactions (Mann & Venkatasubramanian, 2023; Traversa et al., 2023).

Sheaf Hypergraph Networks (SHNs) (Duta et al., 2023) address the first of these challenges by extending the principles of SNNs to hypergraphs. They assign vector spaces to nodes and hyperedges and propagate information via learnable restriction maps naturally mitigating the issues associated with oversmoothing and heterophily while generalizing message passing to higher-order, multi-way interactions, which provides a more expressive framework than traditional HGNNs.

Despite these advantages, SHNs have two key limitations. ($i$) They only model hyperedges as undirected, limiting their ability to capture asymmetric directional relationships. Indeed, while directionality has been recently incorporated in GNNs via, e.g., specialized Laplacians such as the one by Tong et al. (2020) and complex-valued operators (Zhang et al., 2021; Fiorini et al., 2023), extensions to hypergraphs remain limited (Fiorini et al., 2024) and often task-specific (Gatta et al., 2023; Zhao et al., 2024). To our knowledge, no SHN methods which can handle directed hypergraphs are known. $ii$) Second (as we show in this paper), the Laplacian operator proposed in Duta et al. (2023) fails to satisfy the spectral properties required of a well-defined convolutional operator, such as positive semidefiniteness, contrarily to what the authors report (and claim to have proven) in their paper.

In this paper, we introduce *Directional Sheaf Hypergraph Networks* (*DSHN*), a principled extension of SHNs to directed hypergraphs. Specifically, we define *Directed Hypergraph Cellular Sheaves*, equipping hyperedges not only with the notion of tail and head sets (source and target nodes, respectively) but also with *asymmetric* restriction maps that respect orientation within a hyperedge. From this, we derive the *Directed Sheaf Hypergraph Laplacian*, a novel complex-valued Hermitian operator whose phase naturally encodes direction while preserving essential spectral properties, including admitting a spectral decomposition with real-valued, nonnegative eigenvalues. We evaluate DSHN on 7 real-world datasets, as well as on synthetic benchmarks specifically designed to test DSHN's ability to capture directional information within a hypergraph. Compared to 13 state-of-the-art baselines, our method achieves relative accuracy gains from 2% up to 20%, demonstrating that explicitly modeling orientation via our proposed asymmetric and complex-valued restriction maps improves predictive performance. Our contributions can be summarized as follows:

- We introduce the concept of *Directed Hypergraph Cellular Sheaves*, a framework that extends directed hypergraphs by providing a principled representation of directional interactions. This is achieved by assigning complex-valued linear maps between nodes and hyperedges, capturing the node-to-hyperedge relationships within each directed hyperedge.

- We introduce the *Directed Sheaf Hypergraph Laplacian*, a novel complex-valued Hermitian matrix that satisfies the key properties required of a well-defined spectral operator. Our formulation generalizes existing graph and hypergraph Laplacians, providing a unified framework for learning on hypergraphs with both directed and undirected hyperedges.

- We introduce *Directional Sheaf Hypergraph Networks* (DSHN), a model that combines Sheaf theory with a principled treatment of directional information, enabling state-of-the-art performance on directed hypergraph benchmarks.[1]

## 2 BACKGROUND AND PREVIOUS WORK

**Sheaf Neural Networks and Sheaf Hypergraph Networks.** Sheaf theory provides a principled framework for modeling local information flow across structured domains. In algebraic topology, a sheaf associates data to open sets together with restriction maps ensuring consistency (Curry, 2014). Cellular sheaves adapt this idea to cell complexes by assigning vector spaces to cells with linear maps along face relations. Building on this, Sheaf Neural Networks (SNNs) (Hansen & Gebhart, 2020; Bodnar et al., 2022) assign vector spaces to graph nodes and edges and learn restriction maps

---

[1]We provide the code to reproduce the results at `https://github.com/EmaMule/DirectionalSheafHypergraphs`.

for each node-edge incidence relationship, generalizing message passing. SNNs are particularly effective in heterophilic settings and help mitigate oversmoothing, a common drawback of deep GNNs. As described by Hansen & Ghrist (2021), attaching a cellular sheaf to a graph can be interpreted through the lens of opinion dynamics. Unlike traditional Graph Neural Networks (GNNs), each node's representation "lives" in its own vector space, representing a private opinion, while the restriction maps, linear transformations between node and edge vector spaces, govern how this opinion is expressed along each incident edge. In this way, a node can maintain its own distinct representation while still having different "opinions" across the various edges they participate in, allowing for more expressive and flexible representations compared to standard spectral-based methods, where adjacent nodes tend to have similar representations.

Expanding on this idea, Duta et al. (2023) extend SNNs to the hypergraph setting. However, as it will be shown in this work, their formulation suffers from two key limitations. First, it does not capture directionality in hypergraphs. Second, although the *Sheaf Hypergraph Laplacian* is presented as a Laplacian operator, as shown in our paper, it fails to satisfy fundamental spectral properties expected of such operators, most notably positive semidefiniteness.

**Laplacian matrices for Directed Graphs**   Classical spectral methods define convolutional operators through the graph Laplacian (Biggs, 1993; Defferrard et al., 2016b; Kipf & Welling, 2017). While effective, such methods require to either work on inherently undirected graphs or to symmetrize the graph's adjacency matrix, thereby discarding edge directionality. Drawing inspiration from the Magnetic Laplacian introduced by Lieb & Loss in the study of electromagnetic fields, spectral-based methods have been extended to incorporate edge directionality. In particular, Zhang et al. (2021); Fiorini et al. (2023) developed operators that encode orientation in the imaginary part of complex-valued Hermitian matrices. This construction preserves the desirable spectral properties required for a well-defined convolutional operator while embedding directional information, enabling convolutional operators to faithfully capture the asymmetry of directed graphs.

**Undirected and Directed Hypergraphs**   The hyperedge weights are stored in the diagonal matrix $W \in \mathbb{R}^{m \times m}$. The vertex and hyperedge degrees are defined as $\mathbf{D}_u = \sum_{e \in E: u \in e} |w_e|$ for $u \in V$ and $\delta_e = |e|$ for $e \in E$. Hypergraphs where $\delta_e = k$ for some $k \in \mathbb{N}$ for all $e \in E$ are called $k$-uniform. Graphs are 2-uniform hypergraphs. Following Gallo et al. (1993), we define a directed hypergraph as a hypergraph where each edge $e \in E$ is partitioned in a *tail set* $T(e)$ and a *head set* $H(e)$. If $H(e)$ is empty, $e$ is an undirected edge. Research on learning on directed hypergraphs remains limited, with most existing works either constrained to task-specific scenarios (Luo et al., 2022; Gatta et al., 2023) or restricted to 2-uniform directed hypergraphs (Zhao et al., 2024; Ma et al., 2024). This gap has been recently addressed through the Generalized Directed Laplacian (Fiorini et al., 2024), a complex Hermitian operator that unifies directed and undirected hypergraphs, and extends several popular methods for directed graphs to the hypergraph domain. However, their method, while being suitable for directed hypergraphs, is still implicitly biased towards homophilic settings and can be prone to oversmoothing.

## 3   DIRECTED SHEAF HYPERGRAPH LAPLACIAN

### 3.1   DIRECTED HYPERGRAPH CELLULAR SHEAF

In this work, we introduce the notion of *Directed Hypergraph Cellular Sheaf*, which assigns to a directed hypergraph complex-valued restriction maps designed to capture and encode directional information contained in the hypergraph's underlying topology.

**Definition 1.** The Directed Hypergraph Cellular Sheaf of a directed hypergraph $\mathcal{H} = (V, E)$ is the tuple $\langle \mathcal{S}^{(q)}, \{\vec{\mathcal{F}}(u)\}_{u \in V}, \{\vec{\mathcal{F}}(e)\}_{e \in E}, \{\vec{\mathcal{F}}_{u \trianglelefteq e}\}_{u \in \Gamma(e)} \rangle$, consisting of:

1. A complex-valued matrix $\mathcal{S}^{(q)} \in \mathbb{C}^{m \times n}$ with $q \in \mathbb{R}$, defined entry-wise for each hyperedge $e \in E$ and node $u \in V$ as:

$$\mathcal{S}^{(q)}_{u \trianglelefteq e} = \begin{cases} 1 & \text{if } u \in H(e) \quad \text{(head set)} \\ e^{-2\pi i q} & \text{if } u \in T(e) \quad \text{(tail set)} \\ 0 & \text{otherwise} \end{cases}$$

2. A vector space $\vec{\mathcal{F}}(u) \subseteq \mathbb{C}^d$ associated with each node $u \in V$;

3. A vector space $\vec{\mathcal{F}}(e) \subseteq \mathbb{C}^d$ associated with each hyperedge $e \in E$;

4. A restriction map $\vec{\mathcal{F}}_{u \trianglelefteq e} : \vec{\mathcal{F}}(u) \to \vec{\mathcal{F}}(e)$ with $\vec{\mathcal{F}}_{u \trianglelefteq e} = \mathcal{S}_{u \trianglelefteq e}^{(q)} \mathcal{F}_{u \trianglelefteq e} \in \mathbb{C}^{d \times d}$ where $\mathcal{F}_{u \trianglelefteq e} \in \mathbb{R}^{d \times d}$ is a real-valued, directionless, restriction map.

The idea is to associate to each node-hyperedge incidence relationship a linear restriction map $\vec{\mathcal{F}}_{v \trianglelefteq e}$ which can either be real- or complex-valued, based on the directional matrix $\mathcal{S}^{(q)}$, specifying whether a node within a hyperedge belongs to the tail or to the head set. In line with Zhang et al. (2021), although their work focuses on directed graphs and hence fails to model many-to-many interactions, the parameter $q$ associated with the matrix $\mathcal{S}^{(q)}$ serves as a *charge parameter* that controls the relevance of the hypergraph's directional information. In fact, when $q = 0$, the restriction maps are all real-valued independently of the hypergraph's directions, and we come back to the definition of a (Hypergraph) Cellular Sheaf as introduced in Duta et al. (2023). While prior work has incorporated directional information in hypergraphs using complex-valued coefficients, these approaches typically rely on a fixed complex phase (Fiorini et al., 2024). In contrast, our formulation introduces a tunable complex-valued coefficient, allowing the model to flexibly adjust the contribution of directional information. Moreover, by equipping the topology of the hypergraph with a Cellular Sheaf, our method provides a more expressive framework for representing hypergraph structures than existing directed hypergraph neural networks. To visualize of a Directed Hypergraph Cellular Sheaf associated to a directed hyperedge, see Fig. 1.

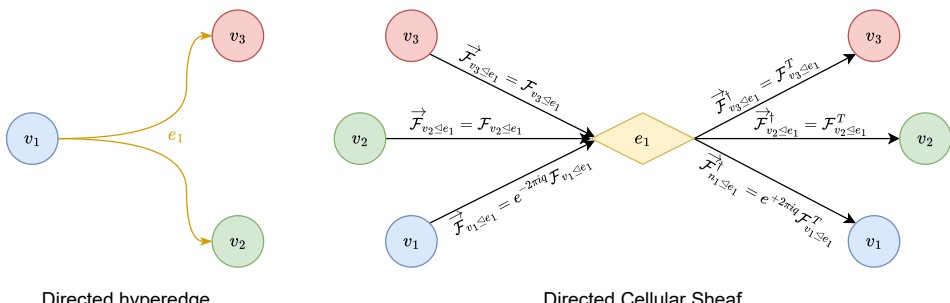

Figure 1: Visualization of sheaves over a directed hyperedge, illustrating the incidence relationship between nodes and the hyperedge, together with the restriction maps $\vec{\mathcal{F}}_{v \trianglelefteq e}$. The tail node $v_1$ is encoded via the $e^{-2\pi i q}$ coefficient which pre-multiplies the directionless restriction map $\mathcal{F}_{v \trianglelefteq e}$.

## 3.2 DIRECTED SHEAF HYPERGRAPH LAPLACIAN

Given a directed hypergraph and its corresponding Directed Hypergraph Cellular Sheaf $\vec{\mathcal{F}}$, let $\mathbf{B}^{(q)} \in \mathbb{C}^{md \times nd}$ be a complex-valued incidence matrix which, for each pair $e \in E$ and $u \in V$, reads:

$$\mathbf{B}_{eu}^{(q)} = \begin{cases} \vec{\mathcal{F}}_{u \trianglelefteq e} = \mathcal{S}_{u \trianglelefteq e}^{(q)} \mathcal{F}_{u \trianglelefteq e} & \text{if } u \in e \\ 0 & \text{otherwise.} \end{cases}$$

When factoring in, for each pair $e \in E$ and $u \in e$, whether $u$ belongs to the head or tail set of $e$, we obtain:

$$\mathbf{B}_{eu}^{(q)} = \begin{cases} \vec{\mathcal{F}}_{u \trianglelefteq e} = \mathcal{S}_{u \trianglelefteq e}^{(q)} \mathcal{F}_{u \trianglelefteq e} = \mathcal{F}_{u \trianglelefteq e} & \text{if } u \in H(e) \quad \text{(head set)} \\ \vec{\mathcal{F}}_{u \trianglelefteq e} = \mathcal{S}_{u \trianglelefteq e}^{(q)} \mathcal{F}_{u \trianglelefteq e} = e^{-2\pi i q} \mathcal{F}_{u \trianglelefteq e} & \text{if } u \in T(e) \quad \text{(tail set)} \\ 0 & \text{otherwise.} \end{cases} \quad (1)$$

We define the *Directed Sheaf Hypergraph Laplacian* $\mathbf{L}^{\vec{\mathcal{F}}}$ associated with a Directed Hypergraph Cellular Sheaf as follows:

$$\mathbf{L}^{\vec{\mathcal{F}}} := \mathbf{D}_V - \mathbf{Q}^{\vec{\mathcal{F}}} \qquad \text{with} \qquad \mathbf{Q}^{\vec{\mathcal{F}}} := \mathbf{B}^{(q)^\dagger} \mathbf{D}_E^{-1} \mathbf{B}^{(q)}, \qquad (2)$$

where $\mathbf{Q}^{\vec{\mathcal{F}}}$ is the *Directed Sheaf Hypergraph Signless Laplacian*.

In the formula, $\mathbf{D}_E$ is the block-diagonal hyperedge degree matrix $\mathbf{D}_E := \mathrm{diag}(\delta_1 \mathbf{I}_d, \dots, \delta_m \mathbf{I}_d) \in \mathbb{R}^{md \times md}$, where $\delta_e := |e|$ is the degree of hyperedge $e \in E$, and $\mathbf{D}_V \in \mathbb{R}^{nd \times nd}$ is the block-diagonal node degree matrix defined as $\mathbf{D}_V := \mathrm{diag}\left(\mathbf{D}_{u_1}, \mathbf{D}_{u_2}, \dots, \mathbf{D}_{u_n}\right)$, with $\mathbf{D}_{u_i} := \sum_{e \in E: u \in e} \vec{\mathcal{F}}^{\dagger}_{u \trianglelefteq e} \vec{\mathcal{F}}_{u \trianglelefteq e} \in \mathbb{R}^{d \times d}, u_i \in V$.

To clarify how $\mathbf{L}^{\vec{\mathcal{F}}}$ encodes the hypergraph structure, let us examine its entry corresponding to a pair of vertices $u, v \in V$:

$$
(\mathbf{L}^{\vec{\mathcal{F}}})_{uv} = \begin{cases} \mathbf{D}_u - \sum_{e:u \in e} \dfrac{1}{\delta_e} \mathcal{F}^{\top}_{u \trianglelefteq e} \mathcal{F}_{u \trianglelefteq e} = \sum_{e:u \in e}(1 - \dfrac{1}{\delta_e}) \mathcal{F}^{\top}_{u \trianglelefteq e} \mathcal{F}_{u \trianglelefteq e} & u = v \\[3mm] -\sum_{e:u,v \in e} \dfrac{1}{\delta_e} \vec{\mathcal{F}}^{\dagger}_{u \trianglelefteq e} \vec{\mathcal{F}}_{v \trianglelefteq e} = -\sum_{e:u,v \in e} \dfrac{1}{\delta_e}(\mathcal{S}^{(q)}_{u \trianglelefteq e})^{\dagger}(\mathcal{S}^{(q)}_{v \trianglelefteq e}) \mathcal{F}^{\top}_{u \trianglelefteq e} \mathcal{F}_{v \trianglelefteq e} & u \neq v. \end{cases}
\tag{3}
$$

Note that the block-diagonal entries of $\mathbf{L}^{\vec{\mathcal{F}}}$ are always real. In contrast, its off-block-diagonal entries are complex-valued when the hypergraph is directed and $q \neq 0$, and real-valued if the hypergraph is undirected. By setting $q = 0$, the hypergraph directions are entirely disregarded and $\mathbf{L}^{\vec{\mathcal{F}}}$ coincides with the Laplacian matrix of its undirected counterpart. The off-diagonal products in $\mathbf{L}^{\vec{\mathcal{F}}}$ strongly depend on the interaction between the two components of the directional matrix $\mathcal{S}^{(q)}$ associated to the restriction maps. The product of these contributes to the real and imaginary parts of $\mathbf{L}^{\vec{\mathcal{F}}}$ as follows:

$$
(\mathcal{S}^{(q)}_{v \trianglelefteq e})^{\dagger} \mathcal{S}^{(q)}_{u \trianglelefteq e} = \begin{cases} 1 & v, u \in T(e) \text{ (tail-tail)}, \\ 1 & v, u \in H(e) \text{ (head-head)}, \\ e^{+2\pi i q} & v \in T(e),\ u \in H(e) \text{ (tail-head)} \quad (\text{equal to } i \text{ if } q = \frac{1}{4}), \\ e^{-2\pi i q} & v \in H(e),\ u \in T(e) \text{ (head-tail)} \quad (\text{equal to } -i \text{ if } q = \frac{1}{4}). \end{cases}
\tag{4}
$$

We can expand Eq. (3) by considering the special case where $q = \frac{1}{4}$. In this case, each entry of $\mathbf{L}^{\vec{\mathcal{F}}}$ can be expressed to explicitly highlight the impact of the directionality of each hyperedge:

$$
(\mathbf{L}^{\vec{\mathcal{F}}})_{uv} = \begin{cases} \displaystyle\sum_{e:u \in e}\left(1 - \frac{1}{\delta_e}\right) \mathcal{F}^{\top}_{u \trianglelefteq e} \mathcal{F}_{u \trianglelefteq e}, & u = v, \\[6mm] -\displaystyle\sum_{\substack{e \in E \\ u,v \in H(e) \\ \vee u,v \in T(e)}} \frac{1}{\delta_e} \mathcal{F}^{\top}_{u \trianglelefteq e} \mathcal{F}_{v \trianglelefteq e} - i\left(\displaystyle\sum_{\substack{e \in E \\ u \in T(e) \\ \wedge v \in H(e)}} \frac{1}{\delta_e} \mathcal{F}^{\top}_{u \trianglelefteq e} \mathcal{F}_{v \trianglelefteq e} - \displaystyle\sum_{\substack{e \in E \\ u \in H(e) \\ \wedge v \in T(e)}} \frac{1}{\delta_e} \mathcal{F}^{\top}_{u \trianglelefteq e} \mathcal{F}_{v \trianglelefteq e}\right), & u \neq v. \end{cases}
\tag{5}
$$

The entry $(\mathbf{L}^{\vec{\mathcal{F}}})_{uv}$ is determined by all hyperedges $e \in E$ that contain both nodes $u$ and $v$. From the second case in Eq. (5), whenever both $u$ and $v$ are both heads ($u, v \in H(e)$) or tails ($u, v \in T(e)$), the contribution to the real part $\Re\left((\mathbf{L}^{\vec{\mathcal{F}}})_{uv}\right)$ is negative and given by the opposite of the normalized weight $-\frac{1}{\delta_e} \mathcal{F}^{\top}_{u \trianglelefteq e} \mathcal{F}_{v \trianglelefteq e}$. In the undirected case, this is the only possible contribution, which matches the expected behavior of undirected hypergraph Sheaf Laplacians, where $u, v \in T(e)$ (or, equivalently, $u, v \in H(e)$). Hyperedges where $u, v$ take opposite roles contribute to the imaginary part with their weight either negatively (if $u \in T(e)$ and $v \in H(e)$) or positively (if $u \in H(e)$ and $v \in T(e)$). Due to this, in the special case where $q = \frac{1}{4}$, $\Im\left((\mathbf{L}^{\vec{\mathcal{F}}})_{uv}\right)$ depends on the net contribution of $u$ and $v$ across all the directed hyperedges that contain them. This is in line with the "net flow" behavior observed in directed graphs (Fiorini et al., 2023) and hypergraphs (Fiorini et al., 2024).

When interpreted as a linear operator acting on a complex signal $\mathbf{x} \in \mathbb{C}^{nd}$, $\mathbf{L}^{\vec{\mathcal{F}}}$ reads:

$$
\left(\mathbf{L}^{\vec{\mathcal{F}}}(\mathbf{x})\right)_u = \sum_{e \in E: u \in e} \frac{1}{\delta_e} \vec{\mathcal{F}}^{\dagger}_{u \trianglelefteq e} \sum_{\substack{v \in e: \\ v \neq u}} \left(\vec{\mathcal{F}}_{u \trianglelefteq e} \mathbf{x}_u - \vec{\mathcal{F}}_{v \trianglelefteq e} \mathbf{x}_v\right).
\tag{6}
$$

We define the *Normalized Directed Sheaf Hypergraph Laplacian* $\mathbf{L}_N^{\vec{\mathcal{F}}}$ as:

$$\mathbf{L}_N^{\vec{\mathcal{F}}} := \mathbf{D}_V^{-\frac{1}{2}} \mathbf{L}^{\vec{\mathcal{F}}} \mathbf{D}_V^{-\frac{1}{2}}.$$

Using Eq. (2), this yields:

$$\mathbf{L}_N^{\vec{\mathcal{F}}} = \mathbf{D}_V^{-\frac{1}{2}} \underbrace{(\mathbf{D}_V - \mathbf{Q}^{\vec{\mathcal{F}}})}_{\mathbf{L}^{\vec{\mathcal{F}}}} \mathbf{D}_V^{-\frac{1}{2}} = \mathbf{I}_{nd} - \mathbf{Q}_N^{\vec{\mathcal{F}}}, \quad \text{where} \quad \mathbf{Q}_N^{\vec{\mathcal{F}}} := \mathbf{D}_V^{-\frac{1}{2}} \mathbf{B}^{(q)\dagger} \mathbf{D}_E^{-1} \mathbf{B}^{(q)} \mathbf{D}_V^{-\frac{1}{2}}. \quad (7)$$

### 3.3 SPECTRAL PROPERTIES

We now establish that our proposed *Normalized Directed Sheaf Hypergraph Laplacian* $\mathbf{L}_N^{\vec{\mathcal{F}}}$ satisfies all the spectral properties required for a principled convolutional operator. Specifically, we show that $\mathbf{L}_N^{\vec{\mathcal{F}}}$ is diagonalizable, has real, nonnegative eigenvalues, is positive semidefinite, and has a bounded spectrum. These ensure that the Fourier transform is well-defined and that polynomial filters of $\mathbf{L}_N^{\vec{\mathcal{F}}}$ implement *localized, stable convolutions*, in direct analogy with classical spectral-based approaches (Shuman et al., 2013; Kipf & Welling, 2017; Defferrard et al., 2016a). Failure to preserve them can result in a complete breakdown of the connection between message passing, the Fourier transform of a (hyper-)graph and its connection with signal theory. The proofs of the claims in this and the next section are provided in Appendix A.

We begin by showing that $\mathbf{L}_N^{\vec{\mathcal{F}}}$ admits an eigenvalue decomposition with real eigenvalues:

**Theorem 1.** $\mathbf{L}_N^{\vec{\mathcal{F}}}$ is diagonalizable with real eigenvalues.

Next, we derive the formula of the Dirichlet energy function associated with $\mathbf{L}_N^{\vec{\mathcal{F}}}$, which provides a measure of the global smoothness of a signal $x \in \mathbb{C}^{nd}$ across the entire hypergraph:

**Theorem 2.** The Dirichlet energy induced by $\mathbf{L}_N^{\vec{\mathcal{F}}}$ for a signal $\mathbf{x} \in \mathbb{C}^{nd}$ is:

$$\mathcal{E}_N(\mathbf{x}) = \mathbf{x}^\dagger \mathbf{L}_N^{\vec{\mathcal{F}}} \mathbf{x} = \frac{1}{2} \sum_{e \in E} \frac{1}{\delta_e} \sum_{\substack{u,v \in e: \\ u \neq v}} \left\| \vec{\mathcal{F}}_{u \unlhd e} \mathbf{D}_u^{-\frac{1}{2}} \mathbf{x}_u - \vec{\mathcal{F}}_{v \unlhd e} \mathbf{D}_v^{-\frac{1}{2}} \mathbf{x}_v \right\|_2^2.$$

By leveraging the two previous theorems, we show that the spectrum of the Directed Sheaf Hypergraph Laplacian $\mathbf{L}_N^{\vec{\mathcal{F}}}$ only admits (real) non-negative eigenvalues:

**Corollary 1.** $\mathbf{L}_N^{\vec{\mathcal{F}}}$ is positive semidefinite.

Finally, we prove that the spectrum of $\mathbf{L}_N^{\vec{\mathcal{F}}}$ is upper bounded by 1:

**Theorem 3.** $\lambda_{\max}(\mathbf{L}_N^{\vec{\mathcal{F}}}) \leq 1$.

### 3.4 GENERALIZATION PROPERTIES

Beyond its spectral properties, our *Directed Sheaf Hypergraph Laplacian* provides a unified definition of a Laplacian matrix that recovers and extends several existing Laplacian matrices.

First, we discuss the relationship between $\mathbf{L}^{\vec{\mathcal{F}}}$ and the *Sheaf Laplacian* introduced by Hansen & Gebhart (2020) and highlight its connection with the classical graph Laplacian defined as $\mathbf{L} := \mathbf{D} - \mathbf{A}$ where $A$ is the graph's adjacency matrix and $D$ its degree matrix (see (Biggs, 1993) for a reference) for the undirected case:

**Theorem 4.** For a 2-uniform hypergraph without directions, the Laplacian operator $\mathbf{L}^{\vec{\mathcal{F}}}$ reduces to the Sheaf Laplacian (Hansen & Gebhart, 2020) (up to a scaling factor of 2) and, when considering the case of a trivial Sheaf (where $\mathcal{F}_{u \unlhd e} = 1$), it coincides with the classical graph Laplacian (up to a scaling factor of 2).

With Theorem 5, we show that $\mathbf{L}^{\vec{\mathcal{F}}}$ generalizes several Laplacians designed for directed graphs like the *Magnetic Laplacian* (Zhang et al., 2021) and the *Sign-Magnetic Laplacian* (Fiorini et al., 2023):

**Theorem 5.** For a directed 2-uniform hypergraph with unitary edge weights (i.e., $w_e = 1, e \in E$) with directed and undirected edges, $\mathbf{L}^{\vec{\mathcal{F}}}$ recovers, as a special case, the Magnetic Laplacian (Zhang et al., 2021) for any $q \in \mathbb{R}$ and the Sign-Magnetic Laplacian (Fiorini et al., 2023) when $q = \frac{1}{4}$.

In the context of hypergraphs, our operator naturally recovers existing hypergraph Laplacians. We begin by showing that it recovers the undirected hypergraph Laplacian of Zhou et al. (2006):

**Theorem 6.** Given a hypergraph $\mathcal{H}$ (directed or undirected), the normalized Directed Hypergraph Laplacian $\mathbf{L}_N^{\vec{\mathcal{F}}}$ recovers, as a special case, the undirected hypergraph Laplacian of Zhou et al. (2006).

We show that $\mathbf{L}^{\vec{\mathcal{F}}}$ generalizes the *Generalized Directed Laplacian* proposed by Fiorini et al. (2024):

**Theorem 7.** Given a directed hypergraph $\mathcal{H}$ with unitary weights associated to each hyperedge (i.e., $w_e = 1$), the Normalized Directed Sheaf Hypergraph Laplacian $\mathbf{L}_N^{\vec{\mathcal{F}}}$ recovers, as a special case, the Generalized Directed Laplacian $\vec{\mathbf{L}}_N$ of Fiorini et al. (2024).

Crucially, our Laplacian operator does not recover the (linear) Sheaf Hypergraph Laplacian proposed in (Duta et al., 2023). This difference is intentional: the formulation proposed in their work, defined for any pair of nodes $u, v \in V$ as:

$$(\mathbf{L}^{\mathcal{F}})_{uu} = \sum_{e:\, u \in e} \frac{1}{\delta_e} \mathcal{F}_{u \trianglelefteq e}^{\top} \mathcal{F}_{u \trianglelefteq e}, \qquad (\mathbf{L}^{\mathcal{F}})_{uv} = - \sum_{\substack{e:u,v \in e \\ v \neq u}} \frac{1}{\delta_e} \mathcal{F}_{u \trianglelefteq e}^{\top} \mathcal{F}_{v \trianglelefteq e}, \qquad (8)$$

fails to satisfy several key spectral properties required to construct a well-defined convolutional operator, most notably positive semidefiniteness. Comparing this definition with Eq. (3), it is evident that, for undirected hypergraphs or when $q = 0$, the two operators differ in the diagonal term: our formulation introduces the factor $\left(1 - \frac{1}{\delta_e}\right)$ rather than $\frac{1}{\delta_e}$. Their definition suffices to guarantee such properties only in the special case of 2-uniform hypergraphs (i.e, standard graphs), however this discrepancy can produce negative eigenvalues for general cases, preventing the matrix from being interpreted as a diffusion operator and precluding the definition of a stable Fourier transform. Polynomial filters based on the Laplacian of Duta et al. (2023) therefore cannot guarantee localized and stable convolutions, limiting its general applicability. A more detailed analysis, including an example that demonstrates this drawback, is presented in Appendix E.

In contrast, when setting $q = 0$ or considering undirected hypergraphs, our operator satisfies all the properties required of a principled Laplacian matrix and, to the best of our knowledge, provides the first definition of a Sheaf Hypergraph Laplacian suitable for undirected hypergraphs.

## 4 DIRECTIONAL SHEAF HYPERGRAPH NETWORK

In this section, we describe our proposed *Directional Sheaf Hypergraph Network* (DSHN). Inspired by Hansen & Gebhart (2020), we define the *sheaf diffusion process* on a directed hypergraph $\mathcal{H}$ as an extension of classical heat diffusion, which plays a central role in spectral-based GNN convolution operators (Kipf & Welling, 2017). Starting from the differential equation:

$$\dot{\mathbf{X}}_t = - \mathbf{L}_N^{\vec{\mathcal{F}}} \mathbf{X_t}.$$

and applying a unit-step Euler discretization, we obtain:

$$\mathbf{X_{t+1}} = \mathbf{X_t} - \mathbf{L}_N^{\vec{\mathcal{F}}} \mathbf{X_t} = (\mathbf{I}_{nd} - \mathbf{L}_N^{\vec{\mathcal{F}}}) \mathbf{X_t}.$$

By introducing learnable parameters and a nonlinear activation function $\sigma$, this discrete diffusion process leads to the following equation of the convolutional layer of DSHN:

$$\mathbf{X}_{t+1} = \sigma\left( (\mathbf{I}_{nd} - \mathbf{L}_N^{\vec{\mathcal{F}}}) (\mathbf{I}_n \otimes \mathbf{W_1}) \mathbf{X}_t \mathbf{W_2} \right) = \sigma\left( \mathbf{Q}_N^{\vec{\mathcal{F}}} (\mathbf{I}_n \otimes \mathbf{W_1}) \mathbf{X}_t \mathbf{W_2} \right) \in \mathbb{C}^{nd \times f}, \quad (9)$$

where $\mathbf{W_1} \in \mathbb{R}^{d \times d}$ and $\mathbf{W_2} \in \mathbb{R}^{f \times f}$ are trainable weight matrices, and $\otimes$ denotes the Kronecker product. $\mathbf{X}_0$ is obtained from the input matrix of node features of size $n \times f$, to which a linear projection is applied to produce an $n \times (df)$ matrix, which is then reshaped into a $(nd) \times f$ matrix before applying the diffusion process. Finally, since the convolutional layer operates in the complex

domain, we transform the output of the final convolutional layer into a real-valued representation right before feeding it to the classification head. Following Zhang et al. (2021); Fiorini et al. (2023), we do so by applying an *unwind* operation, concatenating the real and imaginary components of the complex features as follows:

$$\text{unwind}(\mathbf{X}) = \Re(\mathbf{X}) \,\|\, \Im(\mathbf{X}) \in \mathbb{R}^{n \times 2f}$$

**Restriction Maps** The expressive power of sheaf-theoretical approaches lies in their ability to define a diffusion operator via a learnable $d \times d$ restriction map associated to each node–edge pair. Following Bodnar et al. (2022), we learn each restriction map as a function of the corresponding node and hyperedge features. In particular, for each hyperedge $e \in E$ and each node $u \in e$, each directionless restriction map $\mathcal{F}_{v \trianglelefteq e}$ is parametrized as $\mathcal{F}_{v \trianglelefteq e} = \Phi(\mathbf{x}_v \,\|\, \mathbf{x}_e) \in \mathbb{R}^{d \times d}$, where $x_u$ is the node feature and $\mathbf{x}_e$ is the hyperedge feature and $\Phi$ is an MLP. If hyperedge features are not explicitly provided, they are computed by aggregating the features of the hyperedge's nodes via a mean or a sum. Given that the node and the hyperedge features $\mathbf{x}_v$ and $\mathbf{x}_e$ are complex-valued due to Eq. (9), we employ the same *unwind* operation to map them into a form suitable for input to $\Phi$. Activation functions can be either *sigmoid* or *tanh*.

**DSHNLight** A key objective in our model design is to balance predictive performance with computational efficiency. Constructing an $nd \times nd$ Laplacian matrix from $d \times d$ restriction maps inherently increases complexity, an issue already noted in the graph setting (Bodnar et al., 2022). To mitigate this, we introduce DSHNLight, a variant of DSHN that achieves competitive, and in some cases superior, results across several datasets (Tables 1 and 2) at a significantly lower computational cost (see Appendix C.1). In it, we detach the gradient computation during the Laplacian's construction: this way, the model continues to rely on the predicted restriction maps, but avoids costly gradient propagation. In DSHNLight, the parameters of the MLP responsible for predicting the restriction maps (which are encoded in $\Phi(\cdot)$) remain fixed throughout the training process. The model's adaptability arises from the initial projection layer, which embeds the inputs into a shared feature space where they can be more effectively processed through these parameters (see Appendix C for a better visualization). This phenomenon aligns with insights from the literature on overparameterization (Arora et al., 2019) and extreme learning machines (Huang et al., 2006), where fixed random projections can still yield strong generalization due to the expressive power of the input embeddings. Further details on the difference between the two approaches are discussed in Appendix C.

**Computational Complexity** We provide an estimate of the asymptotic complexity of our model at inference time. Let $n$ denote the number of nodes, $m$ the number of hyperedges, $d$ the stalk dimension, $c$ the product of input and output feature dimensions in the linear transformation, $\bar{e}$ the average hyperedge size, and $\bar{v}$ the average number of hyperedges a node participates in. Summing the contributions from the feature transformation, message passing, learning of restriction maps, and Laplacian assembly, the overall complexity is $\mathcal{O}(n(c^2 + d) + m(\bar{e}d + \bar{e}^2(d + c) + \bar{v}c))$ for diagonal maps, and $\mathcal{O}(n(c^2 + d^3) + m(\bar{e}d^3 + \bar{e}^2(d^3 + dc) + \bar{v}d^2c))$ for non-diagonal maps, sharing the same asymptotic complexity as SheafHyperGNN by Duta et al. (2023). For a comprehensive analysis of the contributions leading to this asymptotic complexity, we refer the reader to Appendix C.1.

## 5 EXPERIMENTAL EVALUATION

We evaluate DSHN and DSHNLight against 13 baseline models from both the directed and undirected hypergraph literature on real-world datasets (Section 5.1) as well as synthetic datasets (Section 5.2) for the node classification task. From the *undirected* hypergraph-learning literature, we include HGNN (Feng et al., 2019), HNHN (Dong et al., 2020), UniGCNII (Huang & Yang, 2021), LEGCN (Yang et al., 2022), HyperND (Tudisco et al., 2021), AllDeepSets and AllSetTransformer (Chien et al., 2022), ED-HNN (Wang et al., 2023a), SheafHyperGNN (Duta et al., 2023), and PhenomNN (Wang et al., 2023b). From the *directed* hypergraph-learning literature, we consider GeDi-HNN (Fiorini et al., 2024) and DHGNN (Ma et al., 2024), along with a variant, as baselines. Model performance is measured in terms of classification accuracy. Following the standard practice in the literature (Chien et al., 2022; Wang et al., 2023a; Fiorini et al., 2024), we adopt a 50%/25%/25% split for training, validation, and testing, respectively, and, for each model, we report the average test accuracy and the standard deviation over 10 independent runs. Details on the baselines, hyperparameter tuning, and the experimental setup are provided in Appendix D.

## 5.1 REAL-WORLD DATASETS

To evaluate our models on real-world datasets, we follow the pre-processing procedure introduced by Tran & Tran (2022) and Fiorini et al. (2024), and apply it to a suite of publicly available directed graph benchmarks to obtain their directed hypergraph counterparts for performing the node classification task (see Appendix D.5). The considered datasets are: `Cora` (Zhang et al., 2022), `email-Enron`, `email-EU` (Benson et al., 2018b), `Telegram` (Bovet & Grindrod, 2020), `Chameleon`, `Squirrel`, and `Roman-empire`. Due to space limitations, Table 1 includes only the datasets that yield the most interesting insights. Additional results can be found in Table 4, while additional informations on the datasets are provided in Appendix D.4.

Table 1: Mean accuracy $\pm$ standard deviation on node classification datasets. For each dataset, the best result is shown in **bold**, and the second best is underlined.

| | Roman-empire | Squirrel | email-EU | Telegram | Chameleon | email-Enron | Cora |
|---|---|---|---|---|---|---|---|
| HGNN | $38.44 \pm 0.44$ | $35.47 \pm 1.44$ | $48.91 \pm 3.11$ | $51.73 \pm 3.38$ | $39.98 \pm 2.28$ | $52.85 \pm 7.27$ | $87.25 \pm 1.01$ |
| HNHN | $46.07 \pm 1.22$ | $35.62 \pm 1.30$ | $29.68 \pm 1.68$ | $38.22 \pm 6.95$ | $35.81 \pm 3.23$ | $18.64 \pm 6.90$ | $78.16 \pm 0.98$ |
| UniGCNII | $78.89 \pm 0.51$ | $38.28 \pm 2.56$ | $44.98 \pm 2.69$ | $51.73 \pm 5.05$ | $39.85 \pm 3.19$ | $47.43 \pm 7.47$ | $87.53 \pm 1.06$ |
| LEGCN | $65.60 \pm 0.41$ | $39.18 \pm 1.54$ | $32.91 \pm 1.83$ | $45.38 \pm 4.23$ | $39.29 \pm 2.04$ | $37.03 \pm 7.16$ | $74.96 \pm 0.94$ |
| HyperND | $68.31 \pm 0.69$ | $40.13 \pm 1.85$ | $32.79 \pm 2.90$ | $44.62 \pm 5.49$ | $44.95 \pm 3.20$ | $38.11 \pm 7.69$ | $78.48 \pm 1.02$ |
| AllDeepSets | $81.79 \pm 0.72$ | $40.69 \pm 1.90$ | $37.37 \pm 6.29$ | $49.19 \pm 6.73$ | $42.97 \pm 3.60$ | $37.29 \pm 7.90$ | $86.86 \pm 0.85$ |
| AllSetTransformer | $83.53 \pm 0.64$ | $40.53 \pm 1.33$ | $38.26 \pm 3.57$ | $66.92 \pm 4.36$ | $43.85 \pm 5.42$ | $63.78 \pm 3.66$ | $86.73 \pm 1.13$ |
| ED-HNN | $83.82 \pm 0.31$ | $39.85 \pm 1.79$ | $68.91 \pm 4.00$ | $60.38 \pm 3.86$ | $44.67 \pm 2.33$ | $51.35 \pm 6.04$ | $86.94 \pm 1.25$ |
| SheafHyperGNN | $74.50 \pm 0.57$ | $42.01 \pm 1.11$ | $52.78 \pm 9.13$ | $70.00 \pm 5.32$ | $41.06 \pm 4.94$ | $63.51 \pm 5.95$ | $87.15 \pm 0.64$ |
| PhenomNN | $71.22 \pm 0.45$ | $39.45 \pm 2.19$ | $37.69 \pm 4.40$ | $47.69 \pm 6.59$ | $43.62 \pm 4.29$ | $47.02 \pm 6.75$ | $\mathbf{88.12 \pm 0.86}$ |
| GeDi-HNN | $83.87 \pm 0.63$ | $43.02 \pm 3.00$ | $52.31 \pm 2.84$ | $77.12 \pm 4.82$ | $39.29 \pm 2.04$ | $50.54 \pm 5.80$ | $85.16 \pm 0.94$ |
| DHGNN | $\underline{77.58 \pm 0.54}$ | $39.85 \pm 1.79$ | $32.35 \pm 2.93$ | $79.62 \pm 5.78$ | $44.08 \pm 4.11$ | $42.16 \pm 8.04$ | $83.16 \pm 1.33$ |
| DHGNN (w/ emb.) | $22.50 \pm 0.81$ | $40.33 \pm 1.42$ | $55.10 \pm 3.48$ | $80.58 \pm 3.89$ | $40.85 \pm 2.76$ | $58.38 \pm 7.57$ | $73.12 \pm 1.04$ |
| **DSHN** | OOM | $\underline{43.55 \pm 2.87}$ | $\underline{78.62 \pm 2.50}$ | $\mathbf{88.65 \pm 5.54}$ | $\mathbf{47.02 \pm 4.35}$ | $\underline{75.68 \pm 3.42}$ | $87.84 \pm 0.90$ |
| **DSHNLight** | $\mathbf{89.24 \pm 0.57}$ | $\mathbf{44.09 \pm 2.36}$ | $\mathbf{82.67 \pm 1.29}$ | $\underline{81.15 \pm 4.19}$ | $\underline{46.50 \pm 4.09}$ | $\mathbf{76.76 \pm 2.48}$ | $\underline{88.02 \pm 1.11}$ |

DSHN, and its variant DSHNLight, which both leverage the theoretical advantages of associating a Directed Cellular Sheaf to a directed hypergraph, consistently outperform the 13 baselines from both the undirected and directed hypergraph learning literature on 6 out of 7 real-world datasets. The largest relative gains are observed on the `email-Enron` and `email-EU` datasets, where DSHN and DSHNLight improve over the best baseline by up to 20%. A substantial improvement is also achieved on `Telegram`, confirming the importance of directional information in this benchmark where all directed methods perform strongly. More moderate but consistent improvements are found on highly heterophilic datasets such as `Roman-empire`, `Chameleon` and `Squirrel` while on highly homophilic datasets such as `Cora` performance is on par with the strongest baselines. As shown in Table 3, the charge parameter $q$ selected by the hyperparameter-selection procedure for our models on highly homophilic datasets is consistently $0.0$. This observation is in line with the findings of Zhang et al. (2021), who report that, in such settings, directional information behaves as noise for node classification.

A better visualization of the impact of the charge parameter $q$ on the predictive performance of our model can be found in Fig. 2, where we highlight the positive impact of directional information on the `Telegram` dataset and how direction is detrimental on the `Cora` dataset. These results not only demonstrate the effectiveness of our models in highly heterophilic settings, but also show how integrating the concept of directionality in hypergraphs can substantially improve performance. Moreover, unlike GeDi-HNN and DHGNN, which are based on Laplacian formulations that embed directionality without any degree of freedom, in our models one can flexibly choose the relevance of directional information by a suitable choice of the charge parameter $q$.

## 5.2 SYNTHETIC DATASETS

We additionally evaluate our models on the synthetic datasets introduced by Fiorini et al. (2024), built over $n = 500$ nodes and split into $c = 5$ classes. Each class contains 30 random intra-class hyperedges, while inter-class directed hyperedges, consisting of multiple tail and head nodes, are added between class pairs with sizes drawn uniformly from $\{3, \ldots, 10\}$. By varying the number of inter-class hyperedges $I_o \in \{10, 30, 50\}$, we control the strength of directional connectivity. This design provides a clean benchmark to test the models' ability to capture directionality; further details are given in Appendix D.4.

Table 2: Mean accuracy $\pm$ standard deviation on the synthetic datasets.

Figure 2: Effect of the charge parameter $q$ on Telegram and Cora.

| Method | $I_o = 10$ | $I_o = 30$ | $I_o = 50$ |
|---|---|---|---|
| HGNN | $47.12 \pm 5.37$ | $43.44 \pm 6.63$ | $37.76 \pm 7.72$ |
| HNHN | $20.40 \pm 2.93$ | $28.88 \pm 9.45$ | $19.76 \pm 3.85$ |
| UniGCNII | $21.44 \pm 4.33$ | $21.12 \pm 2.95$ | $19.84 \pm 2.34$ |
| LEGCN | $17.60 \pm 2.43$ | $20.72 \pm 3.48$ | $19.60 \pm 2.82$ |
| HyperND | $20.40 \pm 2.93$ | $21.12 \pm 3.20$ | $20.64 \pm 1.92$ |
| AllDeepSets | $44.40 \pm 6.81$ | $32.32 \pm 4.82$ | $31.70 \pm 5.92$ |
| AllSetTransformer | $21.12 \pm 3.79$ | $43.68 \pm 8.72$ | $31.84 \pm 3.31$ |
| ED-HNN | $34.00 \pm 6.05$ | $18.88 \pm 2.56$ | $32.48 \pm 6.17$ |
| SheafHyperGNN | $30.64 \pm 5.39$ | $27.28 \pm 7.31$ | $26.00 \pm 9.59$ |
| PhenomNN | $22.24 \pm 4.73$ | $22.08 \pm 4.20$ | $18.72 \pm 3.22$ |
| GeDi-HNN | $71.44 \pm 3.14$ | $71.84 \pm 3.31$ | $78.24 \pm 5.64$ |
| DHGNN | $40.72 \pm 4.55$ | $51.68 \pm 3.97$ | $35.76 \pm 3.70$ |
| DHGNN (w/ emb.) | $84.48 \pm 3.22$ | $85.28 \pm 3.32$ | $81.12 \pm 3.22$ |
| **DSHN** | $94.96 \pm 1.75$ | $\mathbf{97.84 \pm 1.86}$ | $95.84 \pm 2.17$ |
| **DSHNLight** | $\mathbf{95.60 \pm 2.15}$ | $97.04 \pm 2.79$ | $\mathbf{99.04 \pm 0.86}$ |

The results in Table 2 clearly demonstrate the advantage of our models DSHN and DSHNLight over existing baselines. Classical undirected hypergraph methods are unable to capture the directional structure that dominate these benchmarks, and as a result their performance is limited. Directed methods such as GeDi-HNN and DHGNN achieve stronger results, confirming the importance of explicitly incorporating directionality into the convolutional process. Yet, DSHN and DSHNLight, which provide a principled and more expressive treatment of directional structure, yield consistent improvements across all synthetic datasets, outperforming the strongest directed baselines by up to 18 percentage points and reaching 99.04% accuracy on the third synthetic dataset—this highlights the expressive power that the notion of Directed Hypergraph Cellular Sheaves unlocks.

## 6 CONCLUSION AND FUTURE WORKS

We introduced the concept of *Directed Hypergraph Cellular Sheaves* for directed hypergraphs and derived the corresponding *Directed Sheaf Hypergraph Laplacian*, which we integrated into our proposed framework DSHN. By encoding hyperedge direction via a topology-aware complex-valued inductive bias, our method naturally accommodates both directed and undirected hypergraphs while also unifying and generalizing several operators from the graph and hypergraph learning literature. Across a broad set of benchmark datasets, DSHN consistently outperforms methods from both the directed and undirected hypergraph learning literature. As future work, a natural step forward is to evaluate our framework on larger and *natively directed hypergraph datasets* such as protein-protein interaction networks to further test the scalability and expressivity of the method, possibly employing Language Models (LMs) to generate features. Finally, an intriguing direction is to make the *charge parameter $q$* directly learnable, allowing each layer to adapt its diffusion process dynamically.

## ACKNOWLEDGEMENT OF SUPPORT

Antonio Purificato, Federico Siciliano and Fabrizio Silvestri acknowledge projects FAIR (PE0000013), under the MUR National Recovery and Resilience Plan funded by the European Union - NextGenerationEU, and project NEREO (Neural Reasoning over Open Data), funded by the Italian Ministry of Education and Research (PRIN) Grant no. 2022AEFHAZ. Stefano Coniglio's work was partially supported by the European Union under Next Generation EU — the Italian National Recovery and Resilience Plan (PNRR), PRIN 2022 PNRR (project code P20227CTY3, CUP D53D23018800001), project title "HEXAGON: Highly-specialized EXact Algorithms for Grid Operations at the National level".

## REPRODUCIBILITY STATEMENT

We provide all the necessary information to facilitate the reproducibility of our results. Our code repository code can be found here. The README contains all that is needed to set up the Python environment and run the experiments with the different configurations. Further details on the Experimental Setup can be found in Appendix D.

## ETHICS STATEMENT

All datasets employed in this work are publicly available for research and contain no personally identifiable information or harmful content (see Appendix D.4 for further details). The methods introduced in this paper have a societal impact comparable to that of other graph neural networks.

## LLM USAGE STATEMENT

All technical content presented in this paper is entirely our own work, with LLMs serving only as an editorial tool. No scientific content or research findings were generated using an LLM.

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

## A THEORETICAL RESULTS

### A.1 SPECTRAL PROPERTIES

The following Lemma (which we state for clarity even if it is not reported in the paper as a lemma) derives the expression of our proposed Laplacian matrix $\mathbf{L}^{\vec{\mathcal{F}}}$ when applied as a linear operator on a signal:

**Lemma 1.** Let $x \in \mathbb{C}^{nd}$ be a complex-valued signal. Component-wise, the application of $\mathbf{L}^{\vec{\mathcal{F}}}$ to it and to its normalized counterpart reads:

$$\left(\mathbf{L}^{\vec{\mathcal{F}}}(\mathbf{x})\right)_u = \sum_{e:\, u \in e} \frac{1}{\delta_e} \vec{\mathcal{F}}_{u \trianglelefteq e}^{\dagger} \sum_{\substack{v \in e \\ v \neq u}} \left( \vec{\mathcal{F}}_{u \trianglelefteq e}\, \mathbf{x}_u - \vec{\mathcal{F}}_{v \trianglelefteq e}\, \mathbf{x}_v \right).$$

$$\left(\mathbf{L}_N^{\vec{\mathcal{F}}}(\mathbf{x})\right)_u = \sum_{e:\, u \in e} \frac{1}{\delta_e} \left( \mathbf{D}_u^{-\frac{1}{2}} \vec{\mathcal{F}}_{u \trianglelefteq e}^{\dagger} \right) \sum_{\substack{v \in e \\ v \neq u}} \left( \vec{\mathcal{F}}_{u \trianglelefteq e}\, \mathbf{D}_u^{-\frac{1}{2}}\, \mathbf{x}_u - \vec{\mathcal{F}}_{v \trianglelefteq e}\, \mathbf{D}_u^{-\frac{1}{2}}\, \mathbf{x}_v \right).$$

*Proof.* We start by applying the definition of the $\mathbf{L}^{\vec{\mathcal{F}}}$ component-wise as in Eq. (3):

$$
\begin{aligned}
\left(\mathbf{L}^{\vec{\mathcal{F}}}(\mathbf{x})\right)_u &= \sum_{v \in V} (\mathbf{L}^{\vec{\mathcal{F}}})_{uv}\, \mathbf{x}_v \\
&= \sum_{e:\, u \in e} \left( 1 - \frac{1}{\delta_e} \right) \vec{\mathcal{F}}_{u \trianglelefteq e}^{\dagger} \vec{\mathcal{F}}_{u \trianglelefteq e}\, \mathbf{x}_u - \sum_{e:\, u \in e} \sum_{\substack{v \in e \\ v \neq u}} \frac{1}{\delta_e} \vec{\mathcal{F}}_{u \trianglelefteq e}^{\dagger} \vec{\mathcal{F}}_{v \trianglelefteq e}\, \mathbf{x}_v \\
&= \sum_{e:\, u \in e} \frac{1}{\delta_e} \left( (\delta_e - 1) \vec{\mathcal{F}}_{u \trianglelefteq e}^{\dagger} \vec{\mathcal{F}}_{u \trianglelefteq e}\, \mathbf{x}_u - \sum_{\substack{v \in e \\ v \neq u}} \vec{\mathcal{F}}_{u \trianglelefteq e}^{\dagger} \vec{\mathcal{F}}_{v \trianglelefteq e}\, \mathbf{x}_v \right) \\
&= \sum_{e:\, u \in e} \frac{1}{\delta_e} \vec{\mathcal{F}}_{u \trianglelefteq e}^{\dagger} \left( (\delta_e - 1) \vec{\mathcal{F}}_{u \trianglelefteq e}\, \mathbf{x}_u - \sum_{\substack{v \in e \\ v \neq u}} \vec{\mathcal{F}}_{v \trianglelefteq e}\, \mathbf{x}_v \right).
\end{aligned}
$$

Finally, notice that the coefficient $\delta_e - 1$ is exactly the number of vertices in $e$ different from $u$. Thus, the term $(\delta_e - 1) \vec{\mathcal{F}}_{u \trianglelefteq e}\, \mathbf{x}_u$ can be written as a sum of $\vec{\mathcal{F}}_{u \trianglelefteq e}\, \mathbf{x}_u$ over all $v \in e, v \neq u$. Substituting this back, we obtain:

$$
\begin{aligned}
&\sum_{e:\, u \in e} \frac{1}{\delta_e} \vec{\mathcal{F}}_{u \trianglelefteq e}^{\dagger} \left( \sum_{\substack{v \in e \\ v \neq u}} \vec{\mathcal{F}}_{u \trianglelefteq e}\, \mathbf{x}_u - \sum_{\substack{v \in e \\ v \neq u}} \vec{\mathcal{F}}_{v \trianglelefteq e}\, \mathbf{x}_v \right) \\
&= \sum_{e:\, u \in e} \frac{1}{\delta_e} \vec{\mathcal{F}}_{u \trianglelefteq e}^{\dagger} \sum_{\substack{v \in e \\ v \neq u}} \left( \vec{\mathcal{F}}_{u \trianglelefteq e}\, \mathbf{x}_u - \vec{\mathcal{F}}_{v \trianglelefteq e}\, \mathbf{x}_v \right).
\end{aligned}
$$

The linear expression for the normalized case can be derived analogously. $\qquad\square$

In the remainder of this section, we report a proof for each of the theorems we stated in the paper.

**Theorem 1.** $\mathbf{L}_N^{\vec{\mathcal{F}}}$ is diagonalizable with real eigenvalues.

*Proof.* The claim follows rather directly since, as it is not hard to see, $\mathbf{L}_N^{\vec{\mathcal{F}}}$ is Hermitian by construction. $\qquad\square$

**Theorem 2.** The Dirichlet energy induced by $\mathbf{L}_N^{\vec{\mathcal{F}}}$ for a signal $\mathbf{x} \in \mathbb{C}^{nd}$ is:

$$\mathcal{E}_N(\mathbf{x}) = \mathbf{x}^{\dagger} \mathbf{L}_N^{\vec{\mathcal{F}}} \mathbf{x} = \frac{1}{2} \sum_{e \in E} \frac{1}{\delta_e} \sum_{\substack{u,v \in e: \\ u \neq v}} \left\| \vec{\mathcal{F}}_{u \trianglelefteq e}\, \mathbf{D}_u^{-\frac{1}{2}}\, \mathbf{x}_u - \vec{\mathcal{F}}_{v \trianglelefteq e}\, \mathbf{D}_v^{-\frac{1}{2}}\, \mathbf{x}_v \right\|_2^2.$$

*Proof.* By definition of the Dirichlet energy as the quadratic form associated with $\mathbf{L}_N^{\vec{\mathcal{F}}}$, we have:

$$\mathcal{E}_N(\mathbf{x}) = \mathbf{x}^\dagger \mathbf{L}_N^{\vec{\mathcal{F}}} \mathbf{x} = \sum_{u \in V} \mathbf{x}_u^\dagger \big( \mathbf{L}_N^{\vec{\mathcal{F}}}(\mathbf{x}) \big)_u.$$

By substituting for $(\mathbf{L}_N^{\vec{\mathcal{F}}}(\mathbf{x}))_u$ (see the previous lemma), we have:

$$\mathcal{E}_N(\mathbf{x}) = \sum_{u \in V} \sum_{e:\, u \in e} \frac{1}{\delta_e} \sum_{\substack{v \in e \\ v \neq u}} \big( \vec{\mathcal{F}}_{u \unlhd e} \mathbf{D}_u^{-\frac{1}{2}} \mathbf{x}_u \big)^\dagger \left( \vec{\mathcal{F}}_{u \unlhd e}\, \mathbf{D}_u^{-\frac{1}{2}} \mathbf{x}_u - \vec{\mathcal{F}}_{v \unlhd e}\, \mathbf{D}_v^{-\frac{1}{2}} \mathbf{x}_v \right).$$

Distributing the product, we obtain:

$$\mathcal{E}_N(\mathbf{x}) = \sum_{e \in E} \frac{1}{\delta_e} \sum_{u \in e} \sum_{\substack{v \in e \\ v \neq u}} \big( \vec{\mathcal{F}}_{u \unlhd e} \mathbf{D}_u^{-\frac{1}{2}} \mathbf{x}_u \big)^\dagger \vec{\mathcal{F}}_{u \unlhd e} \mathbf{D}_u^{-\frac{1}{2}} \mathbf{x}_u - \sum_{e \in E} \frac{1}{\delta_e} \sum_{\substack{u,v \in e \\ u \neq v}} \big( \vec{\mathcal{F}}_{u \unlhd e} \mathbf{D}_u^{-\frac{1}{2}} \mathbf{x}_u \big)^\dagger \vec{\mathcal{F}}_{v \unlhd e} \mathbf{D}_v^{-\frac{1}{2}} \mathbf{x}_v$$

$$= \sum_{e \in E} \frac{1}{\delta_e} \left( \sum_{u \in e} \sum_{\substack{v \in e \\ v \neq u}} \big\| \vec{\mathcal{F}}_{u \unlhd e} \mathbf{D}_u^{-\frac{1}{2}} \mathbf{x}_u \big\|_2^2 - \sum_{\substack{u,v \in e \\ u \neq v}} \big( \vec{\mathcal{F}}_{u \unlhd e} \mathbf{D}_u^{-\frac{1}{2}} \mathbf{x}_u \big)^\dagger \vec{\mathcal{F}}_{v \unlhd e} \mathbf{D}_v^{-\frac{1}{2}} \mathbf{x}_v \right).$$

Since $\mathbf{L}_N^{\vec{\mathcal{F}}}$ is Hermitian, the second inner summation can be rewritten as:

$$- \sum_{\substack{u,v \in e \\ u \neq v}} \big( \vec{\mathcal{F}}_{u \unlhd e} \mathbf{D}_u^{-\frac{1}{2}} \mathbf{x}_u \big)^\dagger \vec{\mathcal{F}}_{v \unlhd e} \mathbf{D}_v^{-\frac{1}{2}} \mathbf{x}_v =$$

$$- \sum_{\substack{u,v \in e \\ u < v}} \left( \big( \vec{\mathcal{F}}_{u \unlhd e} \mathbf{D}_u^{-\frac{1}{2}} \mathbf{x}_u \big)^\dagger \vec{\mathcal{F}}_{v \unlhd e} \mathbf{D}_v^{-\frac{1}{2}} \mathbf{x}_v + \big( \vec{\mathcal{F}}_{v \unlhd e} \mathbf{D}_v^{-\frac{1}{2}} \mathbf{x}_v \big)^\dagger \vec{\mathcal{F}}_{u \unlhd e} \mathbf{D}_u^{-\frac{1}{2}} \mathbf{x}_u \right) =$$

$$- \sum_{\substack{u,v \in e \\ u < v}} 2 \, \Re \left[ \big( \vec{\mathcal{F}}_{u \unlhd e} \mathbf{D}_u^{-\frac{1}{2}} \mathbf{x}_u \big)^\dagger \vec{\mathcal{F}}_{v \unlhd e} \mathbf{D}_v^{-\frac{1}{2}} \mathbf{x}_v \right] =$$

$$- \sum_{\substack{u,v \in e \\ u \neq v}} \Re \left[ \big( \vec{\mathcal{F}}_{u \unlhd e} \mathbf{D}_u^{-\frac{1}{2}} \mathbf{x}_u \big)^\dagger \vec{\mathcal{F}}_{v \unlhd e} \mathbf{D}_v^{-\frac{1}{2}} \mathbf{x}_v \right].$$

Substituting back and doubling both terms of the summation, we obtain:

$$\mathcal{E}_N(\mathbf{x}) = \frac{1}{2} \sum_{e \in E} \frac{1}{\delta_e} \sum_{\substack{u,v \in e \\ u \neq v}} \left( \big\| \vec{\mathcal{F}}_{u \unlhd e} \mathbf{D}_u^{-\frac{1}{2}} \mathbf{x}_u \big\|_2^2 + \big\| \vec{\mathcal{F}}_{v \unlhd e} \mathbf{D}_v^{-\frac{1}{2}} \mathbf{x}_v \big\|_2^2 - 2 \, \Re \left[ \big( \vec{\mathcal{F}}_{u \unlhd e} \mathbf{D}_u^{-\frac{1}{2}} \mathbf{x}_u \big)^\dagger \vec{\mathcal{F}}_{v \unlhd e} \mathbf{D}_v^{-\frac{1}{2}} \mathbf{x}_v \right] \right).$$

Thanks to the identity $\|a - b\|^2 = \|a\|^2 + \|b\|^2 - 2\Re(a^\dagger b)$, we conclude:

$$\mathcal{E}_N(\mathbf{x}) = \frac{1}{2} \sum_{e \in E} \frac{1}{\delta_e} \sum_{\substack{u,v \in e \\ u \neq v}} \left\| \vec{\mathcal{F}}_{u \unlhd e} \mathbf{D}_u^{-\frac{1}{2}} \mathbf{x}_u - \vec{\mathcal{F}}_{v \unlhd e} \mathbf{D}_v^{-\frac{1}{2}} \mathbf{x}_v \right\|_2^2.$$

Notice that the constraint $u \neq v$ can be dropped from the inner summation w.l.o.g.. $\qquad\square$

**Corollary 1.** $\mathbf{L}_N^{\vec{\mathcal{F}}}$ is positive semidefinite.

*Proof.* This follows directly from the previous theorem. $\qquad\square$

**Theorem 3.** $\lambda_{\max}(\mathbf{L}_N^{\vec{\mathcal{F}}}) \leq 1.$

*Proof.* By definition, we have $\mathbf{L}_N^{\vec{\mathcal{F}}} := \mathbf{I}_{nd} - \mathbf{Q}_N^{\vec{\mathcal{F}}}$, with $\mathbf{Q}_N^{\vec{\mathcal{F}}} := \mathbf{D}_V^{-\frac{1}{2}} \mathbf{B}^{(q)\dagger} \mathbf{D}_E^{-1} \mathbf{B}^{(q)} \mathbf{D}_V^{-\frac{1}{2}}.$

$\mathbf{Q}_N^{\vec{\mathcal{F}}}$ can be factored as

$$\mathbf{Q}_N^{\vec{\mathcal{F}}} = \left(\mathbf{D}_V^{-\frac{1}{2}}\mathbf{B}^{(q)\dagger}\mathbf{D}_E^{-\frac{1}{2}}\right)\left(\mathbf{D}_E^{-\frac{1}{2}}\mathbf{B}^{(q)}\mathbf{D}_V^{-\frac{1}{2}}\right) = \left(\mathbf{D}_E^{-\frac{1}{2}}\mathbf{B}^{(q)}\mathbf{D}_V^{-\frac{1}{2}}\right)^{\dagger}\left(\mathbf{D}_E^{-\frac{1}{2}}\mathbf{B}^{(q)}\mathbf{D}_V^{-\frac{1}{2}}\right).$$

It follows that

$$\mathbf{x}^{\dagger}\mathbf{Q}_N^{\vec{\mathcal{F}}}\mathbf{x} = ||\mathbf{x}^{\dagger}\mathbf{D}_E^{-\frac{1}{2}}\mathbf{B}^{(q)}\mathbf{D}_V^{-\frac{1}{2}}\mathbf{x}||^2 \geq 0,$$

which implies that its spectrum is nonnegative.

Since $\mathbf{L}_N^{\vec{\mathcal{F}}} := \mathbf{I}_{nd} - \mathbf{Q}_N^{\vec{\mathcal{F}}}$, it follows that the spectrum of $\mathbf{L}_N^{\vec{\mathcal{F}}}$ is upper-bounded by 1, which concludes the proof. □

## A.2 GENERALIZATION PROPERTIES

**Theorem 4.** For a 2-uniform hypergraph without directions, the Laplacian operator $\mathbf{L}^{\vec{\mathcal{F}}}$ reduces to the Sheaf Laplacian (Hansen & Gebhart, 2020) (up to a scaling factor of 2) and, when considering the case of a trivial Sheaf (where $\mathcal{F}_{u \trianglelefteq e} = 1$), it coincides with the classical graph Laplacian (up to a scaling factor of 2).

*Proof.* In the 2-uniform case, every hyperedge $e$ contains exactly two nodes (i.e., $\delta_e = 2$). Consider the general expression of the unnormalized Laplacian given in Eq. (3). Since the graph has no directions, $\mathcal{S}_{u \trianglelefteq e}^{(0)} = 1$ for all $u \in V, e \in E$, and for any choice of the charge parameter $q$. As a result, the off-diagonal terms of $\mathbf{L}^{\vec{\mathcal{F}}}$ are real-valued (the diagonal ones always are).

In particular, when $\delta_e = 2$ for all $e \in E$, $\mathbf{L}^{\vec{\mathcal{F}}}$ reads:

$$(\mathbf{L}^{\vec{\mathcal{F}}})_{uv} = \begin{cases} \frac{1}{2} \sum_{e:\, u \in e} \mathcal{F}_{u \trianglelefteq e}^{\top}\mathcal{F}_{u \trianglelefteq e} \in \mathbb{R}^{d \times d}, & u = v, \\ -\frac{1}{2}\, \mathcal{F}_{u \trianglelefteq e}^{\top}\mathcal{F}_{v \trianglelefteq e} \in \mathbb{R}^{d \times d}, & u \neq v, \end{cases}$$

Thus, $\mathbf{L}^{\vec{\mathcal{F}}}$ precisely coincides with the Sheaf Laplacian of Hansen & Gebhart (2020) up to the multiplicative constant $\frac{1}{2}$.

When considering the case of a trivial Sheaf (i.e., when $\mathcal{F}_{v \trianglelefteq e} = 1$), $\mathbf{L}^{\vec{\mathcal{F}}}$ coincides with the definition of the classical graph Laplacian $\mathbf{L} = \mathbf{D} - \mathbf{A}$, where $A$ is the adjacency matrix and $\mathbf{D}$ is the node degree matrix.

Let us note that, in both cases, this constant factor is immaterial in practice, as it can be absorbed by the learnable parameters of the associated neural model. □

**Theorem 5.** For a directed 2-uniform hypergraph with unitary edge weights (i.e., $w_e = 1, e \in E$) with directed and undirected edges, $\mathbf{L}^{\vec{\mathcal{F}}}$ recovers, as a special case, the Magnetic Laplacian (Zhang et al., 2021) for any $q \in \mathbb{R}$ and the Sign-Magnetic Laplacian (Fiorini et al., 2023) when $q = \frac{1}{4}$.

*Proof.* The *Magnetic Laplacian* proposed by Zhang et al. (2021) is defined as

$$\mathbf{L}^{(q)} := \mathbf{D}_s - \mathbf{H}^{(q)} = \mathbf{D}_s - \mathbf{A}_s \odot \exp(i\, \boldsymbol{\Theta}^{(q)}),$$

where $\boldsymbol{\Theta}^{(q)}$ denotes the phase matrix defined as

$$\boldsymbol{\Theta}^{(q)} := 2\pi q\, (\mathbf{A} - \mathbf{A}^{\top})$$

and $\mathbf{A}_s$ is the symmetrized adjacency matrix defined as

$$\mathbf{A}_s := \tfrac{1}{2}(\mathbf{A} + \mathbf{A}^{\top})$$

and $\mathbf{D}_s$ is a diagonal matrix defined as

$$(\mathbf{D}_s)_{uu} := \sum_{v \in V}(\mathbf{A}_s)_{uv} \text{ for all } u \in V.$$

Entry-wise, $\mathbf{H}^{(q)}$ can be written as:

$$\mathbf{H}_{uv}^{(q)} = \begin{cases} \frac{1}{2}e^{2\pi iq} & (u,v) \in E \\ \frac{1}{2}e^{-2\pi iq} & (v,u) \in E \\ 1 & \{u,v\} \in E \\ 0 & \text{otherwise.} \end{cases}$$

In the directed, 2-uniform case, every hyperedge $e$ contains exactly two nodes ($\delta_e = 2$). For every $e \in E$, the product $(\mathcal{S}_{u \trianglelefteq e}^{(q)})^{\dagger} \mathcal{S}_{v \trianglelefteq e}^{(q)}$ can take one of the following three values:

1. Undirected edge $e = \{u,v\}$:

$$\mathcal{S}_{u \trianglelefteq e}^{(q)} = \mathcal{S}_{v \trianglelefteq e}^{(q)} = 1 \implies (\mathcal{S}_{u \trianglelefteq e}^{(q)})^{\dagger} \mathcal{S}_{v \trianglelefteq e}^{(q)} = 1.$$

2. Directed edge $e = (u,v)$:

$$\mathcal{S}_{u \trianglelefteq e}^{(q)} = e^{-2\pi iq}, \qquad \mathcal{S}_{v \trianglelefteq e}^{(q)} = 1 \implies (\mathcal{S}_{u \trianglelefteq e}^{(q)})^{\dagger} \mathcal{S}_{v \trianglelefteq e}^{(q)} = e^{+2\pi iq}.$$

3. Directed edge $e = (v,u)$:

$$\mathcal{S}_{u \trianglelefteq e}^{(q)} = 1, \qquad \mathcal{S}_{v \trianglelefteq e}^{(q)} = e^{-2\pi iq} \implies (\mathcal{S}_{u \trianglelefteq e}^{(q)})^{\dagger} \mathcal{S}_{v \trianglelefteq e}^{(q)} = e^{-2\pi iq}.$$

Letting (w.l.o.g., as the restriction maps are learnable)

$$\begin{cases} \mathcal{F}_{v \trianglelefteq e} := \sqrt{2}, \quad \mathcal{F}_{u \trianglelefteq e} := \sqrt{2} & \text{if } e = \{u,v\}, \\ \mathcal{F}_{v \trianglelefteq e} := 1, \quad \mathcal{F}_{u \trianglelefteq e} = 1 & \text{if } e = (u,v) \text{ or } e = (v,u). \end{cases}$$

we have:

$$(\mathbf{Q}^{\vec{\mathcal{F}}})_{uv} = \begin{cases} \frac{1}{2} \vec{\mathcal{F}}_{u \trianglelefteq e}^{\dagger} \vec{\mathcal{F}}_{v \trianglelefteq e} = \frac{1}{2} e^{+2\pi iq} \mathcal{F}_{u \trianglelefteq e}^{\top} \mathcal{F}_{v \trianglelefteq e} = \frac{1}{2} e^{+2\pi iq}, & \text{if } e = (u,v), \\ \frac{1}{2} \vec{\mathcal{F}}_{u \trianglelefteq e}^{\dagger} \vec{\mathcal{F}}_{v \trianglelefteq e} = \frac{1}{2} e^{-2\pi iq} \mathcal{F}_{u \trianglelefteq e}^{\top} \mathcal{F}_{v \trianglelefteq e} = \frac{1}{2} e^{-2\pi iq}, & \text{if } e = (v,u), \\ \frac{1}{2} \vec{\mathcal{F}}_{u \trianglelefteq e}^{\dagger} \vec{\mathcal{F}}_{v \trianglelefteq e} = \frac{1}{2} \mathcal{F}_{u \trianglelefteq e}^{\top} \mathcal{F}_{v \trianglelefteq e} = 1, & \text{if } e = \{u,v\}. \end{cases}$$

Hence, by construction, we have:

$$\mathbf{Q}^{\vec{\mathcal{F}}} = \mathbf{B}^{(q)^{\dagger}} \mathbf{D}_E^{-1} \mathbf{B}^{(q)} = \mathbf{H}^{(q)}, \text{ with } \mathbf{D}_V = \mathbf{D}_s.$$

This implies:

$$\mathbf{L}^{\vec{\mathcal{F}}} = \mathbf{D}_s - \mathbf{H}^{(q)} = \mathbf{L}^{(q)}.$$

Lastly, noticing that, by construction, the Sign-Magnetic Laplacian proposed in Fiorini et al. (2023) coincides with the Magnetic Laplacian when $q = \frac{1}{4}$, we conclude that our operator also generalizes the former. $\qquad \square$

**Theorem 6.** Given a hypergraph $\mathcal{H}$ (directed or undirected), the normalized Directed Hypergraph Laplacian $\mathbf{L}_N^{\vec{\mathcal{F}}}$ recovers, as a special case, the undirected hypergraph Laplacian of Zhou et al. (2006).

*Proof.* In the unit-weight case, the Laplacian matrix proposed by Zhou et al. (2006) for undirected hypergraphs is defined as follows:

$$\boldsymbol{\Delta} := \mathbf{I} - \mathbf{Q}_N \qquad \text{with } \mathbf{Q}_N := \mathbf{D}_V^{-\frac{1}{2}} \mathbf{B} \mathbf{D}_E^{-1} \mathbf{B}^{\top} \mathbf{D}_V^{-\frac{1}{2}}.$$

Since any undirected hypergraph be regarded as a special case of a directed hypergraph in which every hyperedge consists solely of tail nodes (or, equivalently, solely of head nodes), as shown in Eq. (4), in our proposed Laplacian matrix $\mathbf{L}_N^{\vec{\mathcal{F}}}$ each product of two restriction maps reduces to a real weight of 1, therefore contributing only to the real part of the operator. In particular, for a trivial sheaf where $\mathcal{F}_{v \trianglelefteq e} = 1$, the incidence matrix $\mathbf{B}^{(q)}$ in Eq. (1) reduces to the transpose of binary incidence matrix $B$ of Zhou et al. (2006). $\qquad \square$

**Theorem 7.** Given a directed hypergraph $\mathcal{H}$ with unitary weights associated to each hyperedge (i.e., $w_e = 1$), the Normalized Directed Sheaf Hypergraph Laplacian $\mathbf{L}_N^{\vec{\mathcal{F}}}$ recovers, as a special case, the Generalized Directed Laplacian $\vec{\mathbf{L}}_N$ of Fiorini et al. (2024).

*Proof.* Let's consider a special case of a trivial sheaf (i.e. $\mathcal{F}_{u \trianglelefteq e} = 1$). By setting $q = \frac{1}{4}$ we have:

$$S_{u \trianglelefteq e}^{(0.25)} = \begin{cases} 1 & \text{if } u \in H(e) \quad \text{(head set)} \\ -i & \text{if } u \in T(e) \quad \text{(tail set)} \\ 0 & \text{otherwise} \end{cases}$$

Now, for each pair $u, v$ belonging to the same hyperedge $e$:

$$\vec{\mathcal{F}}_{u \trianglelefteq e}^{\dagger} \vec{\mathcal{F}}_{v \trianglelefteq e} = \left( S_{u \trianglelefteq e}^{(0.25)} \right)^{\dagger} S_{v \trianglelefteq e}^{(0.25)}$$

Whose contribution, according to the four cases in Eq. (4), is given by:

$$\vec{\mathcal{F}}_{u \trianglelefteq e}^{\dagger} \vec{\mathcal{F}}_{v \trianglelefteq e} = \begin{cases} 1, & u, v \in H(e), \\ 1, & u, v \in T(e), \\ i, & u \in T(e), \ v \in H(e), \\ -i, & u \in H(e), \ v \in T(e). \end{cases}$$

Our Normalized Directed Sheaf Hypergraph Laplacian $\mathbf{L}_N^{\vec{\mathcal{F}}}$, component-wise reads:

$$(\mathbf{L}_N^{\vec{\mathcal{F}}})_{uv} = \begin{cases} \mathbf{I}_d - \mathbf{D}_u^{-1} \sum_{e:u \in e} \frac{1}{\delta_e} \mathcal{F}_{u \trianglelefteq e}^{\top} \mathcal{F}_{u \trianglelefteq e} & u = v \\ -\mathbf{D}_u^{-\frac{1}{2}} \Big( \sum_{e:u,v \in e} \frac{1}{\delta_e} \vec{\mathcal{F}}_{u \trianglelefteq e}^{\dagger} \vec{\mathcal{F}}_{v \trianglelefteq e} \Big) \mathbf{D}_v^{-\frac{1}{2}} & u \neq v. \end{cases}$$

Which reduces, in the considered scalar special case to:

$$(\mathbf{L}_N^{\vec{\mathcal{F}}})_{uv} = \begin{cases} 1 - \sum_{e:\, u \in e} \frac{1}{\mathbf{D}_u\, \delta_e}, & u = v, \\ -\sum_{\substack{e \in E \\ u,v \in H(e) \\ \vee\, u,v \in T(e)}} \frac{1}{\delta_e} - i \left( \sum_{\substack{e \in E \\ u \in T(e) \\ \wedge v \in H(e)}} \frac{1}{\delta_e} - \sum_{\substack{e \in E \\ u \in H(e) \\ \wedge v \in T(e)}} \frac{1}{\delta_e} \right) \frac{1}{\sqrt{\mathbf{D}_u}\sqrt{\mathbf{D}_v}}, & u \neq v. \end{cases} \tag{10}$$

Such an expression coincides with the definition of the Generalized Directed Laplacian when considering $\mathbf{W} = \mathbf{I}$. $\square$

## B    EXTENDED EXPERIMENTAL EVALUATION

In this section, we include further experiments and details that did not make the cut in the main paper due to space limits. This includes:

- The optimal value of the charge parameter $q$ found for DSHN and DSHNLight during the hyperparameters optimization process.
- The impact of the stalk dimension $d$ and the number of layers on the method's performance.
- The complete results on 12 real-world datasets.

### B.1    IMPACT OF CHARGE PARAMETER

The charge parameter $q$ controls how much each directed hyperedge contributes to the *real* and *imaginary* parts of the Directed Sheaf Hypergraph Laplacian. Larger values of $q$ place more directional information in the imaginary component, whereas smaller values reduce the directional contribution, emphasizing orientation-agnostic interactions in the real part. Because the dataset differ

in how informative directionality is, the optimal $q$ is inherently data-dependent. In practice, a careful tuning of it is needed $q$ to select the value that yields the best performance, allowing either a partial or a full contribution of directional information to be encoded as needed. Table 3 reports the values chosen by our hyperparameter tuning procedure. As one can seen, for most datasets the hyperparameter tuning procedure sets a relatively high importance to directional information for each dataset, particularly for `Telegram`, `Roman-empire` and Synthetic datasets.

Table 3: Optimal $q$ values for DSHN and DSHNLight across all real-world and synthetic datasets found by hyperparameter tuning.

| Method | Roman-empire | Squirrel | email-EU | Telegram | Chameleon | email-Enron |
|---|---|---|---|---|---|---|
| DSHN | — | 0.05 | 0.25 | 0.25 | 0.20 | 0.05 |
| DSHNLight | 0.20 | 0.05 | 0.20 | 0.20 | 0.15 | 0.15 |

| Method | Cornell | Wisconsin | Amazon-ratings | Texas | Citeseer | Cora |
|---|---|---|---|---|---|---|
| DSHN | 0.25 | 0.25 | — | 0.25 | 0.00 | 0.00 |
| DSHNLight | 0.15 | 0.25 | 0.00 | 0.15 | 0.00 | 0.00 |

| Method | $I_o = 10$ | $I_o = 30$ | $I_o = 50$ |
|---|---|---|---|
| DSHN | 0.25 | 0.10 | 0.10 |
| DSHNLight | 0.10 | 0.10 | 0.10 |

## B.2 IMPACT OF STALK DIMENSION AND NUMBER OF LAYERS

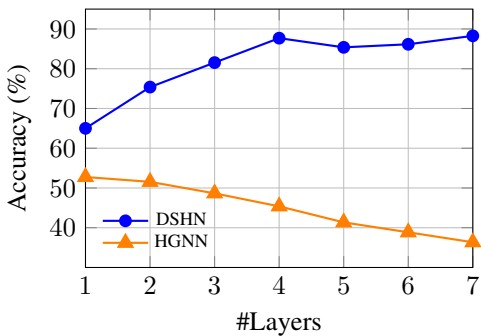 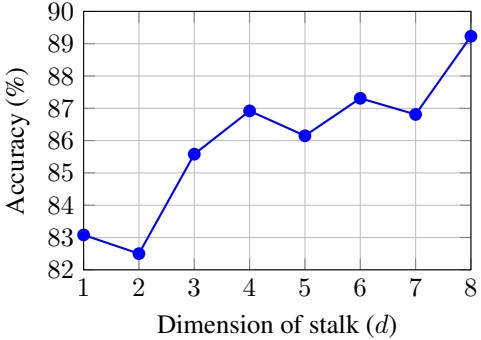

(a) Accuracy of DSHN and HGNN as the number of layers increases.

(b) Accuracy of DSHN as the stalk dimension $d$ increases.

Figure 3: Influence of architectural parameters on accuracy. (a) Effect of the number of layers on DSHN and HGNN. (b) Effect of stalk dimension $d$ on DSHN.

As noted in Section 1, standard HGNNs are prone to *oversmoothing*: as network depth increases, node representations become indistinguishable and accuracy degrades. In Fig. 3, we study how depth and the stalk dimension $d$ affect the accuracy of DSHN. DSHN shows no signs of oversmoothing, as accuracy *improves* as we add layers. Performance also increases with a higher stalk dimension $d$, underscoring the additional expressive power associated to cellular sheaves. This stands in clear contrast to HGNN, whose accuracy steadily deteriorates with depth. This is in line with the observations in Bodnar et al. (2022) for graphs: leveraging our Directed Sheaf Hypergraph Laplacian, built with $d \times d$ restriction maps to transport features between nodes and hyperedges, enriches local variability rather than collapsing it. By projecting node features onto hyperedges (and back), the model retains discriminative power across neighborhoods.

## B.3 EXTENDED RESULTS

In this subsection, we report the complete table of results for this work, which were not reported in the main just due to space limitations. As can be observed from Table 4, DSHN, and DSHNLight consistently outperform the baselines taken from both the directed and undirected hypergraph learning literature on 10 out of 12 considered real-world datasets.

Table 4: Mean accuracy and standard deviation on node classification datasets (test accuracy $\pm$ std). For each dataset, the best result is shown in **bold**, and the second-best is underlined.

| Method | Roman-empire | Squirrel | email-EU | Telegram | Chameleon | email-Enron |
|---|---|---|---|---|---|---|
| HGNN | $38.44 \pm 0.44$ | $35.47 \pm 1.44$ | $48.91 \pm 3.11$ | $51.73 \pm 3.38$ | $39.98 \pm 2.28$ | $52.85 \pm 7.27$ |
| HNHN | $46.07 \pm 1.22$ | $35.62 \pm 1.30$ | $29.68 \pm 1.68$ | $38.22 \pm 6.95$ | $35.81 \pm 3.23$ | $18.64 \pm 6.90$ |
| UniGCNII | $78.89 \pm 0.51$ | $38.28 \pm 2.56$ | $44.98 \pm 2.69$ | $51.73 \pm 5.05$ | $39.85 \pm 3.19$ | $47.43 \pm 7.47$ |
| LEGCN | $65.60 \pm 0.41$ | $39.18 \pm 1.54$ | $32.91 \pm 1.83$ | $45.38 \pm 4.23$ | $39.29 \pm 2.04$ | $37.03 \pm 7.16$ |
| HyperND | $68.31 \pm 0.69$ | $40.13 \pm 1.85$ | $32.79 \pm 2.90$ | $44.62 \pm 5.49$ | $44.95 \pm 3.20$ | $38.11 \pm 7.69$ |
| AllDeepSets | $81.79 \pm 0.72$ | $40.69 \pm 1.90$ | $37.37 \pm 6.29$ | $49.19 \pm 6.73$ | $42.97 \pm 3.60$ | $37.29 \pm 7.90$ |
| AllSetTransformer | $83.53 \pm 0.64$ | $40.53 \pm 1.33$ | $38.26 \pm 3.57$ | $66.92 \pm 4.36$ | $43.85 \pm 5.42$ | $63.78 \pm 3.66$ |
| ED-HNN | $83.82 \pm 0.31$ | $39.85 \pm 1.79$ | $68.91 \pm 4.00$ | $60.38 \pm 3.86$ | $44.67 \pm 2.33$ | $51.35 \pm 6.04$ |
| SheafHyperGNN | $74.50 \pm 0.57$ | $42.01 \pm 1.11$ | $52.78 \pm 9.13$ | $70.00 \pm 5.32$ | $41.06 \pm 4.94$ | $63.51 \pm 5.95$ |
| PhenomNN | $71.22 \pm 0.45$ | $39.45 \pm 2.19$ | $37.69 \pm 4.40$ | $47.69 \pm 6.59$ | $43.62 \pm 4.29$ | $47.02 \pm 6.75$ |
| GeDi-HNN | $\underline{83.87 \pm 0.63}$ | $43.02 \pm 3.00$ | $52.31 \pm 2.84$ | $77.12 \pm 4.82$ | $39.29 \pm 2.04$ | $50.54 \pm 5.80$ |
| DHGNN | $77.58 \pm 0.54$ | $39.85 \pm 1.79$ | $32.35 \pm 2.93$ | $79.62 \pm 5.78$ | $44.08 \pm 4.11$ | $42.16 \pm 8.04$ |
| DHGNN (w/ emb.) | $22.50 \pm 0.81$ | $40.33 \pm 1.42$ | $55.10 \pm 3.48$ | $80.58 \pm 3.89$ | $40.85 \pm 2.76$ | $58.38 \pm 7.57$ |
| **DSHN** | OOM | $\underline{43.55 \pm 2.87}$ | $78.62 \pm 2.50$ | $\mathbf{88.65 \pm 5.54}$ | $\mathbf{47.02 \pm 4.35}$ | $75.68 \pm 3.42$ |
| **DSHNLight** | $\mathbf{89.24 \pm 0.57}$ | $\mathbf{44.09 \pm 2.36}$ | $\mathbf{82.67 \pm 1.29}$ | $81.15 \pm 4.19$ | $46.50 \pm 4.09$ | $\mathbf{76.76 \pm 2.48}$ |

| Method | Cornell | Wisconsin | Amazon-ratings | Texas | Citeseer | Cora |
|---|---|---|---|---|---|---|
| HGNN | $43.51 \pm 6.44$ | $51.56 \pm 6.68$ | $46.20 \pm 0.45$ | $52.77 \pm 7.48$ | $76.02 \pm 0.81$ | $87.25 \pm 1.01$ |
| HNHN | $43.51 \pm 6.09$ | $49.60 \pm 4.96$ | $42.29 \pm 0.34$ | $58.11 \pm 3.87$ | $71.24 \pm 0.66$ | $78.16 \pm 0.98$ |
| UniGCNII | $73.24 \pm 5.19$ | $86.86 \pm 4.30$ | $49.12 \pm 0.46$ | $81.35 \pm 5.33$ | $77.30 \pm 1.15$ | $87.53 \pm 1.06$ |
| LEGCN | $75.14 \pm 5.51$ | $84.71 \pm 4.00$ | $47.02 \pm 0.59$ | $81.35 \pm 4.26$ | $72.62 \pm 1.09$ | $74.96 \pm 0.94$ |
| HyperND | $75.14 \pm 5.38$ | $86.67 \pm 5.02$ | $47.33 \pm 0.51$ | $\underline{83.51 \pm 5.19}$ | $75.21 \pm 1.37$ | $78.48 \pm 1.02$ |
| AllDeepSets | $77.83 \pm 3.78$ | $\underline{87.84 \pm 3.69}$ | $51.91 \pm 0.68$ | $82.76 \pm 5.74$ | $75.78 \pm 0.94$ | $86.86 \pm 0.85$ |
| AllSetTransformer | $75.94 \pm 2.97$ | $86.27 \pm 3.92$ | $52.28 \pm 0.67$ | $82.76 \pm 5.07$ | $75.61 \pm 1.44$ | $86.73 \pm 1.13$ |
| ED-HNN | $76.49 \pm 4.53$ | $85.09 \pm 4.89$ | $51.58 \pm 0.53$ | $80.00 \pm 5.05$ | $74.95 \pm 1.27$ | $86.94 \pm 1.25$ |
| SheafHyperGNN | $74.59 \pm 4.39$ | $85.29 \pm 4.74$ | $48.90 \pm 0.59$ | $80.00 \pm 2.48$ | $77.21 \pm 1.44$ | $87.15 \pm 0.64$ |
| PhenomNN | $72.16 \pm 4.19$ | $80.58 \pm 6.10$ | $48.81 \pm 0.37$ | $81.49 \pm 4.95$ | $77.21 \pm 1.32$ | $\mathbf{88.12 \pm 0.86}$ |
| GeDi-HNN | $\underline{78.37 \pm 3.19}$ | $87.45 \pm 3.41$ | $49.30 \pm 0.52$ | $82.55 \pm 4.64$ | $75.94 \pm 0.95$ | $85.16 \pm 0.94$ |
| DHGNN | $77.30 \pm 4.05$ | $87.45 \pm 3.84$ | $\underline{52.48 \pm 0.50}$ | $83.24 \pm 5.64$ | $74.67 \pm 1.24$ | $83.16 \pm 1.33$ |
| DHGNN (w/ emb.) | $51.08 \pm 4.43$ | $59.80 \pm 5.63$ | $\mathbf{53.64 \pm 0.52}$ | $63.51 \pm 9.84$ | $56.78 \pm 1.32$ | $73.12 \pm 1.04$ |
| **DSHN** | $\mathbf{79.19 \pm 4.37}$ | $\mathbf{88.63 \pm 3.49}$ | OOM | $\mathbf{83.78 \pm 5.13}$ | $77.39 \pm 1.04$ | $87.84 \pm 0.90$ |
| **DSHNLight** | $\mathbf{79.19 \pm 3.20}$ | $87.25 \pm 4.90$ | $50.94 \pm 0.68$ | $82.43 \pm 5.44$ | $\mathbf{77.45 \pm 0.74}$ | $\underline{88.02 \pm 1.11}$ |

Additionally, we evaluate our method on two real-world directed hypergraph dataset for molecular reaction reframed as a hyperedge classification task, results are provided in Table 5. These datasets are the result of the merging of data from different sources such as Kearnes et al. (2021); Reizman et al. (2016); Lugo-Martinez et al. (2021) and are built inspired by Restrepo (2024), which proposes a novel way of modeling molecular reactions through directed hypergraphs. Dataset-1 contains 100,523 nodes and 50,016 hyperedges, with a total of 10 classes. Dataset-2 contains 956 nodes and 3,021 hyperedges to classify among 6 different classes. These datasets consist of inherently directional hyperedges as they contain the molecular reactions expressed as set of reagents (the tail set) and set of products (the head set) composing a molecular reaction. The nodes' features are built based on Morgan Fingerprints (Rogers & Hahn, 2010), which are one of the most widely used molecular descriptors. We employ the F1-score metric since the data has an imbalanced amount of samples for each class as shown in Appendix D.6.

As shown in Table 5, DSHN consistently outperforms all competing methods from both the undirected and directed hypergraph learning literature. On Molecular-1, it achieves an F1-score of 82.32%, improving upon the strongest baseline, GeDi-HNN, by 1.98%. On Molecular-2, DSHN attains 89.09%, exceeding AllSetTransformer by a relative margin of 1.37%.

Table 5: Mean F1-score and standard deviation for hyperedge classification on two molecular reaction datasets (test F1-score $\pm$ std). The best score is shown in **bold**, and the second-best is underlined.

| Method | Molecular-1 | Molecular-2 |
|---|---|---|
| HGNN | $69.38 \pm 0.48$ | $81.40 \pm 2.68$ |
| HNHN | $32.27 \pm 1.30$ | $45.69 \pm 7.48$ |
| UniGCNII | $72.00 \pm 0.59$ | $85.61 \pm 2.63$ |
| LEGCN | OOM | $84.75 \pm 2.68$ |
| HyperND | $44.16 \pm 1.27$ | $82.86 \pm 3.17$ |
| AllDeepSets | $79.17 \pm 0.53$ | $85.78 \pm 3.01$ |
| AllSetTransformer | $79.24 \pm 1.08$ | $\underline{87.89 \pm 2.87}$ |
| ED-HNN | $66.37 \pm 2.62$ | $87.05 \pm 1.96$ |
| SheafHyperGNN | $57.99 \pm 2.75$ | $80.25 \pm 2.20$ |
| PhenomNN | $47.71 \pm 2.90$ | $86.27 \pm 2.40$ |
| GeDi-HNN | $\underline{80.72 \pm 0.78}$ | $85.64 \pm 2.42$ |
| DHGNN | OOM | $85.93 \pm 3.49$ |
| DSHN | OOM | $\mathbf{89.09 \pm 3.08}$ |
| DSHNLight | $\mathbf{82.32 \pm 0.56}$ | $86.52 \pm 2.68$ |

## C  IMPLEMENTATION DETAILS

We provide additional details regarding the implementation of our models, with a particular emphasis on the computational complexity of DSHN and DSHNLight and the architectural choices that contribute to their stability and expressiveness.

### C.1  COMPUTATIONAL COMPLEXITY

**Comparison between DSHN and DSHNLight** Table 6 presents a comparative analysis of DSHN and DSHNLight, across various datasets, measuring their performance in terms of average FLOPS per epoch and average step time. The results are averaged over 10 runs. Over all the 12 datasets, DSHNLight always appears to be more efficient, consistently requiring fewer computational resources while maintaining faster processing times. By applying the aforementioned detachment operation through backpropagation, DSHNLight achieves similar and sometimes better results, as can be seen from Table 1.

Table 6: DSHN vs DSHNLight– FLOPS and Step Time (in ms) Analysis Across Different Datasets (Mean $\pm$ Standard Deviation)

| Dataset | Avg FLOPs/epoch($\downarrow$) | | Avg step time ($\downarrow$) | |
|---|---|---|---|---|
| | DSHN | DSHNLight | DSHN | DSHNLight |
| Cora | $267,070,765,386 \pm 0$ | $196,828,716,921 \pm 3,250$ | $2635.02 \pm 112.51$ | $973.18 \pm 164.18$ |
| Citeseer | $415,705,637,192 \pm 0$ | $310,747,699,339 \pm 5,239$ | $2631.83 \pm 146.54$ | $958.34 \pm 159.00$ |
| email-Enron | $962,025,184 \pm 172$ | $696,069,022 \pm 364$ | $2559.10 \pm 115.97$ | $932.18 \pm 152.09$ |
| email-EU | $35,930,593,693 \pm 1,176$ | $25,798,032,763 \pm 1,183$ | $4170.54 \pm 105.64$ | $1018.13 \pm 155.71$ |
| Telegram | $2,628,033,910 \pm 0$ | $1,858,200,422 \pm 0$ | $2702.33 \pm 150.89$ | $965.40 \pm 161.12$ |
| Cornell | $2,201,584,340 \pm 220$ | $1,851,871,460 \pm 220$ | $2467.84 \pm 132.16$ | $886.07 \pm 164.73$ |
| Texas | $2,228,554,459 \pm 0$ | $1,876,832,187 \pm 0$ | $2480.09 \pm 130.18$ | $888.02 \pm 164.15$ |
| Wisconsin | $3,684,554,183 \pm 201$ | $3,035,436,853 \pm 454$ | $2547.95 \pm 116.24$ | $923.18 \pm 161.33$ |
| Chameleon | $34,986,115,734 \pm 123$ | $27,033,570,342 \pm 123$ | $2629.45 \pm 132.52$ | $959.08 \pm 155.75$ |
| Squirrel | $189,607,210,489 \pm 5,694$ | $140,531,198,787 \pm 3,557$ | $3870.93 \pm 119.79$ | $1046.00 \pm 169.09$ |
| Roman-empire | OOM | $12,898,147,996,391 \pm 43,606$ | OOM | $1050.30 \pm 152.04$ |
| Amazon-ratings | OOM | $15,061,770,374,298 \pm 0$ | OOM | $1080.26 \pm 159.91$ |

**Comparison between DSHN and other models** Fig. 4 reports the average test accuracy of five representative models under approximately the same parameter budget. The results indicate that model size alone does not explain the performance of DSHN. For instance, although SheafHyper-GNN and ED-HNN have a comparable number of parameters, their accuracy is significantly lower, being these undirected methods. In contrast, DSHN achieves an improvement of about 8% over the

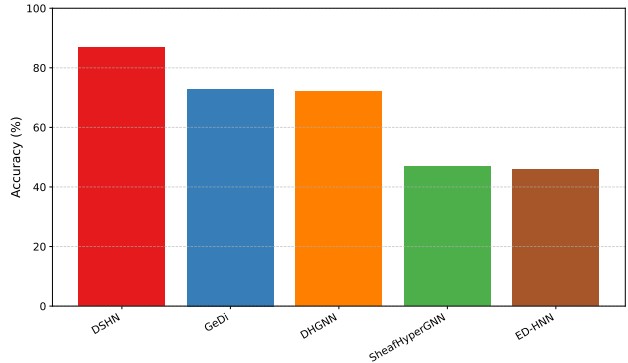

Figure 4: Comparison between models under the same number of parameters ($\sim 80k$) on the Telegram dataset.

strongest directed baselines, despite having the same number of parameters thanks to the expressive power associated to complex-valued and directional restriction maps. On the other hand, Table 7 shows a comparison of DSHNLight with other models. Both DSHNLight and SheafHyperGNN generally incur higher computational costs than traditional hypergraph neural networks, although achieving, at least in the case of DSHNLight, a substantially better accuracy. This overhead stems directly from the requirement to learn and apply restriction maps at every node-hyperedge incidence as detailed in the next paragraph.

Table 7: FLOPs and Parameter Count Across Datasets and Methods.

| Dataset | ED-HNN | | SheafHyperGNN | | DHGNN | | DSHNLight | |
|---|---|---|---|---|---|---|---|---|
| | FLOPs | #Params | FLOPs | #Params | FLOPs | #Params | FLOPs | #Params |
| Cora | $3 \times 10^9$ | 125,959 | $2 \times 10^{11}$ | 404,576 | $7 \times 10^{10}$ | 4,077,051 | $2.0 \times 10^{11}$ | 437,447 |
| Citeseer | $5 \times 10^9$ | 271,174 | $3 \times 10^{11}$ | 985,440 | $3 \times 10^{11}$ | 12,751,318 | $3.1 \times 10^{11}$ | 1,018,502 |
| email-Enron | $4 \times 10^8$ | 34,311 | $6 \times 10^8$ | 37,984 | $2 \times 10^7$ | 13,334 | $7.0 \times 10^8$ | 70,855 |
| email-EU | $5 \times 10^9$ | 34,506 | $3 \times 10^{10}$ | 38,752 | $5 \times 10^8$ | 14,372 | $2.6 \times 10^{10}$ | 71,050 |
| Telegram | $1 \times 10^9$ | 34,116 | $2 \times 10^9$ | 37,216 | $4 \times 10^7$ | 13,241 | $1.9 \times 10^9$ | 70,660 |
| Cornell | $2 \times 10^8$ | 143,109 | $1 \times 10^9$ | 473,184 | $6 \times 10^8$ | 542,566 | $1.9 \times 10^9$ | 506,437 |
| Texas | $2 \times 10^8$ | 143,109 | $1 \times 10^9$ | 473,184 | $6 \times 10^8$ | 542,566 | $1.9 \times 10^9$ | 506,437 |
| Wisconsin | $3 \times 10^8$ | 143,109 | $2 \times 10^9$ | 473,184 | $1 \times 10^9$ | 658,370 | $3.0 \times 10^9$ | 506,437 |
| Chameleon | $2 \times 10^9$ | 182,917 | $2 \times 10^{10}$ | 632,416 | $1 \times 10^{10}$ | 2,379,783 | $2.7 \times 10^{10}$ | 632,416 |
| Squirrel | $9 \times 10^9$ | 167,813 | $1 \times 10^{11}$ | 572,000 | $7 \times 10^{10}$ | 4,924,172 | $1.3 \times 10^{11}$ | 605,253 |
| Roman-empire | $2 \times 10^{10}$ | 54,162 | $1 \times 10^{13}$ | 117,344 | $1 \times 10^{12}$ | 6,850,778 | $1.3 \times 10^{13}$ | 117,344 |
| Amazon-ratings | $3 \times 10^{10}$ | 53,317 | $2 \times 10^{13}$ | 114,016 | $1 \times 10^{12}$ | 7,398,933 | $1.5 \times 10^{13}$ | 114,016 |

**Asymptotic Complexity**   We provide an estimate of the asymptotic complexity of our model at inference time.

1. **Linear Transformation.** The feature transformation is defined as

$$\mathbf{X}' = (\mathbf{I}_n \otimes \mathbf{W}_1)\mathbf{X}\mathbf{W}_2$$

where $\mathbf{W}_1 \in \mathbb{R}^{d \times d}$ and $\mathbf{W}_2 \in \mathbb{R}^{f \times f}$. The resulting complexity is $\mathcal{O}(n(d^2 f + df^2)) = \mathcal{O}(n(cd + cf)) = \mathcal{O}(nc^2)$, where $c = df$.

2. **Message Passing.** Once the Laplacian operator has been assembled, message passing reduces to a sparse-dense matrix multiplication of the form

$$\mathbf{Q}_N^{\vec{\mathcal{F}}}\mathbf{X}'.$$

The sparsity pattern of $\mathbf{Q}_N^{\vec{\mathcal{F}}}$ comes directly from the incidence matrix: each hyperedge of size $|e|$ induces $|e|^2$ nonzero blocks through the outer product $\mathbf{B}^{(q)}(e,:)^\dagger \mathbf{B}^{(q)}(e,:)$. Summing across all hyperedges gives a total of $\mathsf{S}_2 = \sum_{e \in \mathcal{E}} |e|^2 = \mathcal{O}(m\bar{e}^2)$ nonzero blocks,

where $\bar{e}$ is the average hyperedge size[2]. Applying the Laplacian then requires $\mathcal{O}(m\bar{e}^2 c)$ for diagonal maps and $\mathcal{O}(m\bar{e}^2 dc)$ with non-diagonal maps,

3. **Learning the Sheaf.** Restriction maps are predicted as

$$\Phi(\mathbf{x}_v, \mathbf{x}_e) = \sigma\big(\mathbf{V}\big(\mathbf{x}_v \,\|\, \mathbf{x}_e\big)\big)$$

where $\mathbf{V}$ is a learnable transformation and $\sigma$ a nonlinearity. The resulting $f$-dimensional vector is then used as input to $V$ for every node-edge incidence. Indicating with $\bar{v}$ the average number of partecipations of a node to an hyperedge, the computational complexity is $\mathcal{O}(\bar{v}mc)$ in the *diagonal* case, and $\mathcal{O}(\bar{v}md^2c)$ in the *non-diagonal* case.

4. **Constructing the Laplacian.** In the hypergraph setting we assemble

$$\mathbf{Q}_N^{\vec{\mathcal{F}}} = \mathbf{D}_V^{-\frac{1}{2}}\mathbf{B}^{(q)^\dagger}\mathbf{D}_E^{-1}\mathbf{B}^{(q)}\mathbf{D}_V^{-\frac{1}{2}}.$$

The work naturally splits into two steps:

(a) *Degree normalization.* This involves computing the node and hyperedge degree matrices, $\mathbf{D}_V^{-\frac{1}{2}}$ and $\mathbf{D}_E^{-1}$. For vertex degree normalization each node requires aggregating contributions from its incident hyperedges, giving $\mathcal{O}(m\bar{e}d)$ operations in the diagonal case and $\mathcal{O}(m\bar{e}d^3)$ in the non-diagonal case (since each block is $d \times d$), to which it must be added the cost of inverting the block-diagonal matrices, adding to the complexity $\mathcal{O}(nd)$ in the diagonal case and $\mathcal{O}(nd^3)$ in the non-diagonal case while since $\mathbf{D}_E^{-1}$ is obtained by expanding to matrix for the scalar hyperedge degrees $\delta_e$ this cost adds a trascurable term to the asymptotic complexity.

(b) *Sparse product.* Forming the term

$$\mathbf{Q}_N^{\vec{\mathcal{F}}} = \mathbf{D}_V^{-\frac{1}{2}}\mathbf{B}^{(q)^\dagger}\mathbf{D}_E^{-1}\mathbf{B}^{(q)}\mathbf{D}_V^{-\frac{1}{2}}.$$

requires, for each hyperedge $e$, generating block interactions among all pairs of nodes it contains. This gives a total of $\mathsf{S}_2 = \sum_{e \in \mathcal{E}} |e|^2$ block products. The cost is $\mathcal{O}(\mathsf{S}_2 d)$ in the diagonal case and $\mathcal{O}(\mathsf{S}_2 d^3)$ in the non-diagonal case. Since the normalization terms $\mathbf{D}_V^{-\frac{1}{2}}$ are block-diagonal operations they do not contribute substantially in the overall complexity. Since $\mathsf{S}_2 = \mathcal{O}(m\bar{e}^2)$, the dominant cost becomes $\mathcal{O}(m\bar{e}^2 d)$ for diagonal maps and $\mathcal{O}(m\bar{e}^2 d^3)$ for non-diagonal maps.

By summing the overall contributions we get: $\mathcal{O}\big(n\,(c^2 + d)\ +\ m\,(\bar{e}d + \bar{e}^2(d+c) + \bar{v}\,c)\big)$ in the diagonal case and $\mathcal{O}\big(n\,(c^2 + d^3) + m\,(\bar{e}d^3 + \bar{e}^2(d^3 + dc) + \bar{v}\,d^2c)\big)$ in the non-diagonal case.

**Considerations on the Asymptotic Complexity**  The leading cost arises from the *Laplacian assembly step*, which scales as $\mathcal{O}(m\bar{e}^2 d)$ in the diagonal case and $\mathcal{O}(m\bar{e}^2 d^3)$ in the non-diagonal case. This quadratic dependence on the average hyperedge size $\bar{e}^2$ makes the method particularly sensitive to hypergraphs with densely populated hyperedges. In practice, this means that even when the number of nodes and hyperedges are moderate, the presence of densely populated hyperedges can dominate the computational cost.

## C.2 ARCHITECTURAL CHOICES

**Layer Normalization**  Each layer may optionally include layer normalization, with this choice considered a tunable hyperparameter, since it improves training stability and overall performance. Since the input signal to each convolutional layer is complex-valued, we adopt a complex normalization strategy as proposed in Trabelsi et al. (2018); Barrachina et al. (2023), where each complex feature is treated as a two-dimensional real vector $(\Re(x), \Im(x))$. Specifically, we compute the full $2 \times 2$ covariance matrix:

$$\Sigma = \begin{bmatrix} \sigma_{rr} & \sigma_{ri} \\ \sigma_{ri} & \sigma_{ii} \end{bmatrix}, \quad \tilde{\mathbf{x}} = \Sigma^{-\frac{1}{2}}(\mathbf{x} - \boldsymbol{\mu}),$$

---

[2]One could also upper bound the $\mathsf{S}_2$ term with $\mathcal{O}(mn^2)$, however, that approximation would be highly pessimistic, considering a fully-dense representation of the hypergraph, where each hyperedge connects all nodes.

where $\boldsymbol{\mu} = (\mu_r, \mu_i)$ is the mean vector of the real and imaginary parts. The whitening transform $\Sigma^{-\frac{1}{2}}$ ensures that the two components are jointly normalized and decorrelated. To enhance flexibility, we apply an optional learnable affine transformation in the complex plane:

$$x_o = \gamma \tilde{x} + \beta,$$

with trainable parameters $\gamma \in \mathbb{R}^{2 \times 2}$ and $\beta \in \mathbb{R}^2$. These are initialized as $\gamma = \frac{1}{\sqrt{2}} I_2$ and $\beta = 0$, thereby preserving the norm of unit-modulus inputs while maintaining the identity mapping at initialization.

**Residual Connections**  Following observations from Bodnar et al. (2022), we optionally include residual connections in our convolutional layers, which we found to help the architecture in certain datasets. The use of residuals is treated as a tunable hyperparameter (see Appendix D.3). With this addition, a convolutional layer takes the form:

$$\mathbf{X}_{t+1} = \sigma \left( \mathbf{Q}_N^{\vec{\mathcal{F}}} \left( \mathbf{I}_n \otimes \mathbf{W}_1 \right) \mathbf{X}_t \, \mathbf{W}_2 + \mathbf{X}_t \right) \in \mathbb{C}^{nd \times f}.$$

**Activation Function**  For the activation function, we adopt the complex ReLU commonly employed in related works (Zhang et al., 2021; Fiorini et al., 2023; 2024). It is defined as:

$$\text{ReLU}(x) = \begin{cases} x, & \text{if } \Re(x) > 0, \\ 0, & \text{otherwise.} \end{cases}$$

**DSHNLight**  The architecture of DSHNLight is illustrated in Fig. 5. The model takes as input a node feature matrix $X_{\text{input}}$, which is projected into a higher-dimensional stalk space via a learnable linear transformation. This representation is then used both in the message-passing pipeline and as input to the MLP that predicts the restriction maps $\vec{\mathcal{F}}_{v \trianglelefteq e}$. Unlike DSHN, the Laplacian operator is built outside the computational graph, so the MLP parameters are not updated during training. Nevertheless, the initial projection layer remains trainable, which allows the model to indirectly influence the restriction maps: by shaping the input embeddings, the network can still control the outputs of the MLP. In this way, even though the restriction map MLPs are frozen, the model is still able to predict good values of embeddings and restriction maps, as confirmed by the empirical results in Tables 2 and 4.

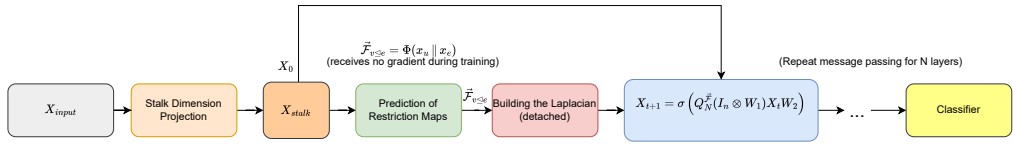

Figure 5: Illustration of the DSHNLight architecture. The Laplacian construction is detached from the computational graph, but the initial stalk projection layer remains trainable, allowing the model to indirectly influence the restriction maps.

# D  EXPERIMENTAL SETUP

## D.1  HARDWARE DETAILS

All experiments are carried out on two different workstations: one equipped with two NVIDIA RTX 4090 GPUs (24 GB each) and an AMD Ryzen 9 7950X 6-core processor, and another featuring an Intel Core i9-10940X 14-core CPU (3.3 GHz), 256 GB of RAM, and a single NVIDIA RTX A6000 GPU with 48 GB of VRAM. We utilized the WandB platform to monitor training procedures and to carry out hyperparameter tuning for each model.

## D.2  SELECTED BASELINES

We compare our models against twelve state-of-the-art methods from the hypergraph learning literature. From the *undirected* hypergraph-learning literature we include HGNN (Feng et al., 2019),

HNHN (Dong et al., 2020), UniGCNII (Huang & Yang, 2021), LEGCN (Yang et al., 2022), HyperND (Tudisco et al., 2021), AllDeepSets and AllSetTransformer (Chien et al., 2022), ED-HNN (Wang et al., 2023a), SheafHyperGNN (Duta et al., 2023) and PhenomNN (Wang et al., 2023b). From the *directed* hypergraph-learning literature, we consider GeDi-HNN (Fiorini et al., 2024) and DHGNN (Ma et al., 2024) as our baselines. DHGNN was originally designed for link prediction on directed graph datasets and relies on a learnable embedding table to represent node features. In our evaluation, we report the model's performance using both this original embedding approach and an alternative setup with explicit node features.

## D.3 HYPERPARAMETER TUNING

For tuning all the models, we employ a Bayesian optimization method. All models are trained for up to 500 epochs with early stopping set to 200 epochs. We employ Adam (Kingma & Ba, 2017) for optimizing the model parameters with $\text{lr} \in \{0.02, 0.01, 0.005, 0.001\}$, $\text{wd} \in \{0, 5 \times 10^{-5}, 5 \times 10^{-4}\}$. For all the models, we adopt a dropout $\in \{0.1, 0.2, \dots, 0.9\}$, and for each model that has a selectable number of layers for the final classifier we fix it to 2. For each baseline, we select a range of parameters consistent with those investigated in their respective original works:

- AllDeepSets, ED-HNN: basic blocks $\{2, 4, 8\}$; MLPs per block $\{1, 2\}$; MLP hidden width $\{64, 128, 256, 512\}$; classifier width $\{64, 128, 256\}$.
- AllSetTransformer: basic blocks $\{2, 4, 8\}$; MLPs per block $\{1, 2\}$; hidden MLP width $\{64, 128, 256, 512\}$; classifier width $\{64, 128, 256\}$; heads $\{1, 4, 8\}$.
- UniGCNII, HGNN, HNHN, LEGCN: basic blocks $\{2, 4, 8\}$; MLP hidden width $\{64, 128, 256, 512\}$.
- HyperND: classifier width $\{64, 128, 256\}$.
- PhenomNN: basic blocks $\{2, 4, 8\}$; hidden width $\{64, 128, 256, 512\}$; $\lambda_0 \in \{0.1, 0, 1\}$; $\lambda_1 \in \{0.1, 50, 1, 20\}$; propagation steps $\{8, 16\}$.
- GeDi-HNN: convolutional layers $\{1, 2, 3\}$; MLP hidden width $\{64, 128, 256, 512\}$; classifier width $\{64, 128, 256\}$.
- DHGNN, DHGNN (w/ emb.), basic blocks $\{2, 4, 8\}$; hidden width $\{64, 128, 256, 512\}$, classifier width $\{64, 128, 256\}$.
- SheafHyperGNN, DSHN, DSHNLight:
  - sheaf dropout $\in \{\texttt{false}, \texttt{true}\}$
  - convolutional layers $\in \{1, \dots, 5\}$
  - MLP hidden width $\{64, 128, 256, 512\}$
  - classifier width $\{64, 128, 256\}$
  - $d \in \{1, \dots, 6\}$
  - sheaf actvation $\in \{\texttt{sigmoid}, \texttt{tanh}, \texttt{none}\}$
  - left projection $\in \{\texttt{false}, \texttt{true}\}$
  - residual $\in \{\texttt{false}, \texttt{true}\}$
  - dynamic sheaf $\in \{\texttt{false}, \texttt{true}\}$
  - $q \in \{0.00, 0.05, 0.10, 0.15, 0.20, 0.25\}$ (for DSHN & DSHNLight only)

## D.4 DATASETS DESCRIPTION

We follow the data splits proposed by Zhang et al. (2021) for the `Telegram`, `Texas`, `Wisconsin`, `Cornell`. For `Chameleon` and `Squirrel` we adopt the splits proposed by Platonov et al. (2023). For `Roman-empire` and `Amazon-Ratings` we adopt the splits proposed by Platonov et al. (2023) and adopt the splits of Chien et al. (2022) for the remaining ones. In all cases, the datasets are partitioned into 50% training, 25% validation, and 25% test samples. For the `email-Enron` and `email-EU` datasets and for all synthetic datasets, node attributes are not available. In these cases, we resort to structural features, representing each node by its degree. The statistics of the 12 real-world datasets as well as synthetic ones are provided in Table 8. The datasets used for the experiments are:

- `Cora`, `Citeseer` Standard citation benchmarks in which vertices represent research papers and directed edges encode citation links. Node attributes are constructed from text using bag-of-words representations of the documents.

- `email-Enron`, `email-EU` A corporate email communication network built from Enron's message logs. Nodes correspond to email accounts and edges record sender interactions. As ground-truth labels are unavailable, we derive node classes via the Spinglass community detection method Reichardt & Bornholdt (2006).

- `Texas`, `Wisconsin`, `Cornell` WebKB datasets collected from university computer science departments. Each node is a webpage, hyperlinks are edges, and features are bag-of-words over page content. Pages are annotated into five categories: student, project, course, staff, and faculty.

- `Telegram` An interaction network extracted from Telegram, capturing exchanges among users who propagate political content.

- `Squirrel`, `Chameleon` The Squirrel and Chameleon datasets consist of articles from the English Wikipedia (December 2018). Nodes represent articles, and edges represent mutual links between them. Node features indicate the presence of specific nouns in the articles. Nodes are grouped into five categories based on the original regression targets.

- `Roman-empire` The dataset is based on the *Roman Empire* article from English Wikipedia, which was selected since it is one of the longest articles on Wikipedia and it follows the construction proposed by Platonov et al. (2023). Each node in the graph corresponds to one (non-unique) word in the text.

- `Amazon-ratings` The dataset, as proposed by Platonov et al. (2023), is based on the Amazon product co-purchasing network metadata dataset from SNAP Datasets Leskovec & Krevl (2014). Nodes are products (books, music CDs, DVDs, VHS video tapes), and edges connect products that are frequently bought together.

- `Synthetic` Introduced in Fiorini et al. (2024) by following the methodology adopted in Zhang et al. (2021), these datasets are built as follows: a vertex set $V$ is partitioned into $c$ equally sized classes $C_1, \ldots, C_c$. For each class $C_i$, we sample $I_i$ *intra-class* hyperedges that are undirected. The cardinality of each hyperedge is drawn uniformly from $\{h_{\min}, \ldots, h_{\max}\}$, and its nodes are sampled uniformly from $C_i$. For each ordered pair of distinct classes $(C_i, C_j)$ with $i < j$, we create $I_o$ *inter-class directed* hyperedges. For every such hyperedge $e$, the tail set $T(e)$ is sampled from $C_i$ and the head set $H(e)$ from $C_j$; the sizes $|T(e)|$ and $|H(e)|$ are drawn uniformly from $\{h_{\min}, \ldots, h_{\max}\}$.

Table 8: Statistics of the datasets used in our experiments. Reported are the number of nodes, features, hyperedges, and classes, as well as the average hyperedge size ($|e|$), the average node degree ($|v|$), and the clique-expansion (CE) homophily computed as in Wang et al. (2023a).

| Dataset | # Nodes | # Features | # Hyperedges | # Classes | avg $|e|$ | avg $|v|$ | CE homophily |
|---|---|---|---|---|---|---|---|
| Roman-empire | 22,662 | 300 | 22,662 | 18 | 2.73 | 2.73 | 0.2363 |
| Squirrel | 2,223 | 2,089 | 2,060 | 5 | 23.81 | 22.07 | 0.2448 |
| email-EU | 986 | – | 787 | 10 | 43.36 | 34.61 | 0.2608 |
| Telegram | 245 | 1 | 183 | 4 | 49.70 | 37.12 | 0.2854 |
| Chameleon | 890 | 2,325 | 797 | 5 | 12.11 | 10.84 | 0.3221 |
| email-Enron | 143 | – | 139 | 7 | 19.58 | 19.03 | 0.3251 |
| Cornell | 183 | 1,703 | 96 | 5 | 4.07 | 2.14 | 0.4200 |
| Wisconsin | 251 | 1,703 | 170 | 5 | 3.94 | 2.67 | 0.4398 |
| Amazon-ratings | 24,492 | 300 | 24,456 | 5 | 5.63 | 5.62 | 0.4460 |
| Texas | 183 | 1,703 | 110 | 5 | 3.81 | 2.29 | 0.5049 |
| Citeseer | 3,312 | 3,703 | 1,951 | 6 | 3.35 | 1.98 | 0.7947 |
| Cora | 2,708 | 1,433 | 1,565 | 7 | 4.47 | 2.58 | 0.8035 |
| $I_o = 10$ | 500 | – | 250 | 5 | 9.05 | 4.53 | 0.6233 |
| $I_o = 30$ | 500 | – | 450 | 5 | 10.79 | 9.71 | 0.5020 |
| $I_o = 50$ | 500 | – | 650 | 5 | 11.63 | 15.12 | 0.4528 |

### D.5 DIRECTED HYPERGRAPH FROM A DIRECTED GRAPH

Given a directed graph $G = (V, E)$, let the out-neighborhood of $v \in V$ be

$$N_{\text{out}}(v) = \{\, w \in V \mid (v, w) \in E \,\}.$$

We build a directed hypergraph $\mathcal{H} = (V, \mathcal{E})$ by creating one hyperedge $e_v$ for each node with its outgoing edges and setting

$$T(e_v) = \{v\}, \qquad H(e_v) = N_{\text{out}}(v).$$

Thus every hyperedge has a tail consisting of a single node and a head set containing all nodes belonging to the neighborhood of that tail. A clear example of this construction procedure can be visualized in Fig. 6.

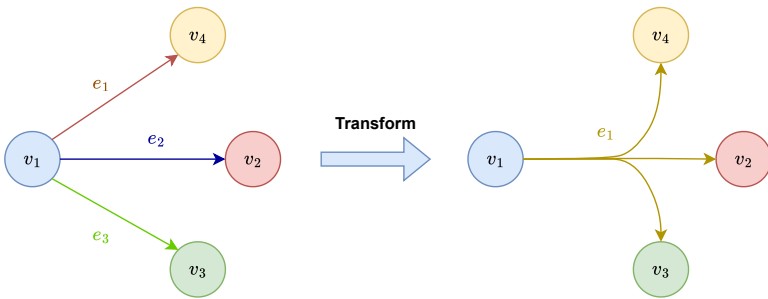

Figure 6: Example of the creation of a directed hyperedge from the out-neighborhood of a node. Suppose we have a graph where node $v_1$ connects to nodes $v_2$, $v_3$, and $v_4$, so that $(v_1, v_2)$, $(v_1, v_3)$, and $(v_1, v_4)$ belong to $E$. The construction procedure yields a directed hyperedge $e_1$ with tail set $T(e_1) = \{v_1\}$ and head set $H(e_1) = \{v_2, v_3, v_4\}$.

This formulation preserves the source-target semantics of the original graph by expressing them as a higher-order relation. Such hyperedges are often referred to as *forward directed* hyperedges (Gallo et al., 1993). When every hyperedge is forward directed, the structure is a *forward directed hypergraph*, which is the case for all real-world datasets considered in this work.

### D.6 HYPEREDGE CLASSIFICATION FOR MOLECULAR REACTION TYPE PREDICTION

In Table 9 and Table 10 we show the distribution of labels for the directed molecular reaction prediction datasets employed for the hyperedge classification task. As mentioned in Appendix B.3, we evaluate all models on additional real-world molecular prediction datasets (see Table 5) from the perspective of a hyperedge classification task. To do so, before feeding the output of the last convolutional layer to the classifier, we perform an aggregation (sum) of all node representations belonging to a given hyperedge. Specifically, if $\mathbf{X}_{\text{node}}$ is the final feature matrix of shape $N \times F$, where $N$ is the number of nodes and $F$ is the feature dimension, we can compute hyperedge-level representations using a (real-valued and binary) incidence matrix $\mathbf{H}$ of shape $N \times E$ as follows:

$$\mathbf{H}^\top \mathbf{X}_{\text{node}} = \mathbf{X}_{\text{edge}} \in \mathbb{R}^{E \times F}.$$

The resulting matrix $\mathbf{X}_{\text{edge}}$ is then fed to the classifier which will output a matrix of shape $E \times C$, where $C$ is the number of classes.

## E ON PREVIOUS PROPOSALS OF THE SHEAF HYPERGRAPH LAPLACIAN

In this section, we revisit the definition of the Sheaf Hypergraph Laplacian proposed in Duta et al. (2023), noting that it fails to satisfy basic spectral properties expected of a Laplacian operator, most notably positive semidefiniteness. This shortcoming motivates our formulation, which, as discussed in Section 3.4, constitutes (to our knowledge) the first definition of a Sheaf Hypergraph Laplacian that is fully consistent with the spectral requirements of a convolutional operator also in the undirected setting. For comparison, we recall the (called linear in the paper—the nonlinear one is, in essence, the Laplacian of a 2-uniform hypergraph) Laplacian of Duta et al. (2023).

| Class | # Hyperedges | Percentage (%) |
|-------|--------------|----------------|
| 0 | 15,151 | 30.29 |
| 1 | 11,896 | 23.78 |
| 2 | 5,662 | 11.32 |
| 3 | 909 | 1.82 |
| 4 | 672 | 1.34 |
| 5 | 8,237 | 16.47 |
| 6 | 4,614 | 9.23 |
| 7 | 811 | 1.62 |
| 8 | 1,834 | 3.67 |
| 9 | 230 | 0.46 |

Table 9: Label distribution for the Molecular-1 dataset.

| Class | # Hyperedges | Percentage (%) |
|-------|--------------|----------------|
| 0 | 960 | 31.78 |
| 1 | 1,536 | 50.84 |
| 2 | 213 | 7.05 |
| 3 | 54 | 1.79 |
| 4 | 226 | 7.48 |
| 5 | 32 | 1.06 |

Table 10: Label distribution for the Molecular-2 dataset.

**Definition 2.** Let $\mathcal{H} = (V, E)$ be a hypergraph with hyperedge degrees $\delta_e$ and let $\mathcal{F}_{v \trianglelefteq e} : \mathbb{R}^d \to \mathbb{R}^d$ be linear restriction maps from node $v$ to hyperedge $e$. The Laplacian $\mathbf{L}^{\mathcal{F}} \in \mathbb{R}^{nd \times nd}$ has $d \times d$ blocks indexed by $u, v \in V$:

$$(\mathbf{L}^{\mathcal{F}})_{uu} = \sum_{e:\, u \in e} \frac{1}{\delta_e} \mathcal{F}_{u \trianglelefteq e}^{\top} \mathcal{F}_{u \trianglelefteq e}, \qquad (\mathbf{L}^{\mathcal{F}})_{uv} = -\sum_{\substack{e:\, u, v \in e \\ v \neq u}} \frac{1}{\delta_e} \mathcal{F}_{u \trianglelefteq e}^{\top} \mathcal{F}_{v \trianglelefteq e}$$

Definition 2 essentially coincides with a Signless Hypergraph Laplacian, except for the fact that the off-diagonal entries are flipped from positive to negative.[3] Such a sign-flip suffices to build a positive semidefinite Laplacian matrix exclusively in the 2-uniform case, where the Laplacian operator for a graph can be obtained by assigning an arbitrary orientation to each edge. Notice that, in the undirected case, our Laplacian differs from theirs due to featuring a coefficient of $(1 - \frac{1}{\delta_e})$ in the diagonal term, rather than $\frac{1}{\delta_e}$. Considering the proposed definition, we can compute the equation of the Laplacian seen as a linear operator for a signal $x \in \mathbb{R}^{nd}$ as follows:

$$\left(\mathbf{L}^{\mathcal{F}}(\mathbf{x})\right)_u = \sum_{v \in V} (\mathbf{L}^{\mathcal{F}})_{uv}\, \mathbf{x}_v$$

$$= \sum_{e:\, u \in e} \frac{1}{\delta_e} \mathcal{F}_{u \trianglelefteq e}^{\top} \mathcal{F}_{u \trianglelefteq e}\, \mathbf{x}_u - \sum_{e:\, u \in e} \sum_{\substack{v \in e \\ v \neq u}} \frac{1}{\delta_e} \mathcal{F}_{u \trianglelefteq e}^{\top} \mathcal{F}_{v \trianglelefteq e}\, \mathbf{x}_v$$

$$= \sum_{e:\, u \in e} \frac{1}{\delta_e} \left( \mathcal{F}_{u \trianglelefteq e}^{\top} \mathcal{F}_{u \trianglelefteq e}\, \mathbf{x}_u - \sum_{\substack{v \in e \\ v \neq u}} \mathcal{F}_{u \trianglelefteq e}^{\top} \mathcal{F}_{v \trianglelefteq e}\, \mathbf{x}_v \right)$$

$$= \sum_{e:\, u \in e} \frac{1}{\delta_e} \mathcal{F}_{u \trianglelefteq e}^{\top} \left( \mathcal{F}_{u \trianglelefteq e}\, \mathbf{x}_u - \sum_{\substack{v \in e \\ v \neq u}} \mathcal{F}_{v \trianglelefteq e}\, \mathbf{x}_v \right).$$

---

[3]This is consistent with their implementation.

Which substantially differs from the expression reported in their respective work, which reads:

$$\left(\mathbf{L}^{\mathcal{F}}(\mathbf{x})\right)_u = \sum_{e:\, u \in e} \frac{1}{\delta_e} \mathcal{F}_{u \trianglelefteq e}^{\top} \sum_{\substack{v \in e \\ v \neq u}} \left(\mathcal{F}_{u \trianglelefteq e}\, \mathbf{x}_u - \mathcal{F}_{v \trianglelefteq e}\, \mathbf{x}_v\right).$$

Crucially, the latter is the expression that is obtained with our operator in the undirected case, as reported in Eq. (6).

Let us illustrate the issue with a numerical example. Let us consider a hypergraph with node set $V = \{v_1, v_2, v_3, v_4\}$ and $E = \{e_1, e_2\}$ with hyperedges $e_1 = \{v_1, v_2, v_3\}$, $e_2 = \{v_2, v_3, v_4\}$, in the case of a *trivial* Sheaf (i.e. $\mathcal{F}_{v \trianglelefteq e} = 1$). Let $\delta_e$ denote the hyperedge size and let $\mathcal{F}_{u \trianglelefteq e} \in \mathbb{R}$ be the (scalar) restriction on incidence $(u, e)$.

By Definition 2, the entries of the Laplacian are:

$$(\mathbf{L}^{\mathcal{F}})_{v_1 v_1} = \tfrac{1}{\delta_{e_1}} \mathcal{F}_{v_1 \trianglelefteq e_1}^{\top} \mathcal{F}_{v_1 \trianglelefteq e_1},$$

$$(\mathbf{L}^{\mathcal{F}})_{v_2 v_2} = \tfrac{1}{\delta_{e_1}} \mathcal{F}_{v_2 \trianglelefteq e_1}^{\top} \mathcal{F}_{v_2 \trianglelefteq e_1} + \tfrac{1}{\delta_{e_2}} \mathcal{F}_{v_2 \trianglelefteq e_2}^{\top} \mathcal{F}_{v_2 \trianglelefteq e_2},$$

$$(\mathbf{L}^{\mathcal{F}})_{v_3 v_3} = \tfrac{1}{\delta_{e_1}} \mathcal{F}_{v_3 \trianglelefteq e_1}^{\top} \mathcal{F}_{v_3 \trianglelefteq e_1} + \tfrac{1}{\delta_{e_2}} \mathcal{F}_{v_3 \trianglelefteq e_2}^{\top} \mathcal{F}_{v_3 \trianglelefteq e_2},$$

$$(\mathbf{L}^{\mathcal{F}})_{v_4 v_4} = \tfrac{1}{\delta_{e_2}} \mathcal{F}_{v_4 \trianglelefteq e_2}^{\top} \mathcal{F}_{v_4 \trianglelefteq e_2},$$

$$(\mathbf{L}^{\mathcal{F}})_{v_1 v_2} = -\tfrac{1}{\delta_{e_1}} \mathcal{F}_{v_1 \trianglelefteq e_1}^{\top} \mathcal{F}_{v_2 \trianglelefteq e_1},$$

$$(\mathbf{L}^{\mathcal{F}})_{v_1 v_3} = -\tfrac{1}{\delta_{e_1}} \mathcal{F}_{v_1 \trianglelefteq e_1}^{\top} \mathcal{F}_{v_3 \trianglelefteq e_1},$$

$$(\mathbf{L}^{\mathcal{F}})_{v_1 v_4} = 0,$$

$$(\mathbf{L}^{\mathcal{F}})_{v_2 v_3} = -\tfrac{1}{\delta_{e_1}} \mathcal{F}_{v_2 \trianglelefteq e_1}^{\top} \mathcal{F}_{v_3 \trianglelefteq e_1} - \tfrac{1}{\delta_{e_2}} \mathcal{F}_{v_2 \trianglelefteq e_2}^{\top} \mathcal{F}_{v_3 \trianglelefteq e_2},$$

$$(\mathbf{L}^{\mathcal{F}})_{v_2 v_4} = -\tfrac{1}{\delta_{e_2}} \mathcal{F}_{v_2 \trianglelefteq e_2}^{\top} \mathcal{F}_{v_4 \trianglelefteq e_2},$$

$$(\mathbf{L}^{\mathcal{F}})_{v_3 v_4} = -\tfrac{1}{\delta_{e_2}} \mathcal{F}_{v_3 \trianglelefteq e_2}^{\top} \mathcal{F}_{v_4 \trianglelefteq e_2}.$$

Numerically, we have:

$$\begin{bmatrix} \frac{1}{3} & -\frac{1}{3} & -\frac{1}{3} & 0 \\ -\frac{1}{3} & \frac{2}{3} & -\frac{2}{3} & -\frac{1}{3} \\ -\frac{1}{3} & -\frac{2}{3} & \frac{2}{3} & -\frac{1}{3} \\ 0 & -\frac{1}{3} & -\frac{1}{3} & \frac{1}{3} \end{bmatrix}.$$

The spectrum of the above Laplacian is:

$$\operatorname{eig}\left(\mathbf{L}^{\mathcal{F}}\right) = \left\{ \tfrac{4}{3},\ \tfrac{1}{3},\ \tfrac{1+\sqrt{17}}{6},\ \tfrac{1-\sqrt{17}}{6} \right\}.$$

Since a negative eigenvalue appears, $\mathbf{L}^{\mathcal{F}}$ is *not* positive semidefinite in this example.

