# OpenReview forum: "Directional Sheaf Hypergraph Networks: Unifying Learning on Directed and Undirected Hypergraphs"
_ICLR.cc/2026/Conference — ICLR 2026 Poster_

### Official Review · Reviewer_YUAy · 2025-10-23

**Soundness:** 3
**Presentation:** 2
**Contribution:** 2
**Rating:** 6
**Confidence:** 4

**Summary:**

The paper addresses the absence of a directed Sheaf Hypergraph Laplacian in the literature by proposing a novel definition, together with the introduction of corresponding Directed Hypergraph Cellular Sheaves and a Directional Sheaf Hypergraph Network. The authors motivate their approach by noting that existing work fails to properly address node classification in directed hypergraphs, particularly under heterophilic conditions. They evaluate the proposed model on hypergraph datasets with varying levels of homophily and further assess its effectiveness in a synthetic experimental setting.

**Strengths:**

- The work addresses a problem that was not thoroughly investigated in the literature before, providing a theoretically interesting solution.
- The proposed method is well justified theoretically, with several proofs regarding the properties of the newly defined Laplacian.
- The motivations, goal and method are overall clear.
- The study on the effects of the parameter $q$ in Fig. 2 provides a nice and easily understandable justification for its introduction.

**Weaknesses:**

A major concern is the lack of an introduction to the Sheaf Hypergraph Network (SHN) [1] method, along with a comparison to your proposed method, in the main text. Strong claims are made in the introduction, stating that SHN does not actually possess some of the properties claimed by its authors, and that the proposed approach addresses this issue. However, in the main text there is no direct reference - even at an intuitive level - to these claims or to how the new method compares to SHN.

Additionally, the introduction of a sheaf structure generally adds computational and runtime overhead compared to a simple prediction model, so its inclusion should be well justified in two respects: (1) whether the additional cost is outweighed by the gain in prediction accuracy, and (2) whether there is a real practical need for a sheaf-based model. Regarding (1), a discussion of the computational cost, ideally comparing the model with non-sheaf baselines, is missing. Regarding (2), I refer to the absence of experiments on natively directed hypergraph datasets, since the considered ones are artificially derived from existing directed graph benchmarks. I understand that this limitation may be difficult to address experimentally due to time constraints, but it would still be helpful to provide some discussion about the availability of such datasets and whether they have been used in similar experimental settings, to clarify the data landscape and the actual necessity for directional hypergraph methods.

[1] Duta et al., Sheaf hypergaph networks, 2023

**Questions:**

**General concerns**
- Section 3.3 provides solid theoretical results, but two elements are missing: (1) a discussion of the practical meaning of these properties, how they relate to the model’s behavior when used to define a convolutional architecture, and (if possible) a connection to the experimental results; and (2) a comparison to SHN, as mentioned above, to explicitly support your claims.
- Could you be more specific regarding the novelty of the objects you use in your definitions? For example, it is not clear to me whether the object $\mathcal{S}^{(q)}$ represents a novel contribution or not.
- What is the computational and runtime overhead associated to your model with respect to SHN and other non-sheaf-based hypergraph networks?
- In the experimental section, there is no introduction or citation of some of the most relevant baselines (e.g., GeDi-HNN, DHGNN).

**Structure of the paper**
- Looking at the overall structure, I would argue that too little space is dedicated to properly introducing the background and previous methods that are central to the work (e.g., SHN), while a large portion focuses on detailed derivations of various sheaf objects. I suggest moving some of these explicit equations to the Appendix and keeping more intuitive explanations in the main text.

**Notation**
- I recommend to define more thoroughly the notation used in Section 2, for example specify how $| . |$ is used in different contexts, and unify the notations $\delta_e$ and $\delta(e)$.

---

> ### Author Response · Authors · 2025-11-20
> **Response to Reviewer YUAy - Part 1**
>
> >A major concern is the lack of an introduction to the Sheaf Hypergraph Network (SHN) [1] method, along with a comparison to your proposed method, in the main text. Strong claims are made in the introduction, stating that SHN does not actually possess some of the properties claimed by its authors, and that the proposed approach addresses this issue. However, in the main text there is no direct reference - even at an intuitive level - to these claims or to how the new method compares to SHN.
>
> In the revised manuscript, we have added a detailed discussion at the end of Section 3 that directly addresses this issue. This new text provides an intuitive explanation of the key differences between our approach and SHN, and clearly explains why SHN does not satisfy certain desirable properties that our method guarantees. We believe this addition makes our contribution clearer. All changes are highlighted in blue in the updated PDF to facilitate your review.
>
> >(1) whether the additional cost is outweighed by the gain in prediction accuracy, and (2) whether there is a real practical need for a sheaf-based model. Regarding (1), a discussion of the computational cost, ideally comparing the model with non-sheaf baselines, is missing. Regarding (2), I refer to the absence of experiments on natively directed hypergraph datasets, since the considered ones are artificially derived from existing directed graph benchmarks.
>
> Regarding point (1), in appendix C.1 we provide several experiments concerning the computational complexity of both variants of our model, including an asymptotic complexity analysis of the two at inference time, which we now also report in the main paper. Moreover, Figure 4 from the Appendix investigates the computational cost concern. This figure shows that when comparing models with approximately the same parameter budget, our approach consistently outperforms all baselines. Model size alone does not explain the performance gains of our approach. For instance, SheafHyperGNN and ED-HNN have a comparable number of parameters to our method in that case, yet their accuracy is significantly lower, a difference we attribute to the fact that these are undirected methods that cannot properly capture the directional structure of the data. This suggests that our Directed Hypergraph Cellular Sheaf is not simply adding overhead, but is actually enabling the model to better capture the underlying data structure. Following your suggestions we have, however, also included an additional comparison of our models against other sheaf and non-sheaf baselines in terms of FLOPs which you can find in Table 7 in the supplementary material.
>
> Regarding point (2), to address this limitation, we have now included experiments on two natively directed hypergraph datasets. We believe this addition better demonstrates the practical applicability of our method to real-world directed hypergraph data. See the answer to reviewer **rWeC** (Part 1) for more information.

---

> > ### Author Response · Authors · 2025-11-20
> > **Response to Reviewer YUAy - Part 2**
> >
> > >(1) a discussion of the practical meaning of these properties, how they relate to the model’s behavior when used to define a convolutional architecture, and (if possible) a connection to the experimental results; (2) a comparison to SHN, as mentioned above, to explicitly support your claims.
> >
> > The properties we prove are not merely formal requirements but they ensure that the Fourier transform is well-defined and that polynomial filters of $\mathbf{L}^{\vec{\mathcal{F}}}_N$ implement localized, stable convolutions, in direct analogy with classical spectral-based approaches. Failure to preserve these properties can result in a complete breakdown of the connection between message passing, the Fourier transform of a (hyper-)graph and its connections with signal theory. Without these guarantees, we cannot reliably interpret or control the behavior of the network. We have added a dedicated explanation in Section 3.3.
> >
> > To better understand this idea, we provide Table 1. This table compares our DSHN method with SheafHyperGNN [1] in the special case of $q=0$. As highlighted in the paper, this setting corresponds to our Laplacian effectively ignoring directional information, thus aligning with the non-directional context considered by [1].
> >
> > Table 1: Node classification accuracy (%) comparison between DSHN (with directional information disregarded, q=0) and SheafHyperGNN
> > | Method       | Roman       | Squirrel     | email-EU     | Telegram     | Chameleon    | email-Enron  | Cora        |
> > |-------------|------------|-------------|-------------|-------------|-------------|-------------|------------|
> > | DSHN (q=0)  | **83.58 $\pm$ 0.49** | 41.62 $\pm$ 1.74 | **77.04 $\pm$ 1.44** | **75.00 $\pm$ 6.49** | 40.49 $\pm$ 3.03 | **67.84 $\pm$ 4.26** | **87.84 $\pm$ 0.90** |
> > | SheafHyperGNN        | 74.50 $\pm$ 0.57 | **42.01 $\pm$ 1.11** | 52.78 $\pm$ 9.13 | 70.00 $\pm$ 5.32 | **41.06 $\pm$ 4.94** | 63.51 $\pm$ 5.95 | 87.15 $\pm$ 0.64 |
> >
> > *For Roman-empire, we report the accuracy of DSHNLight.
> >
> > As shown in Table 1, our method remains generally more expressive and stable than the approach of SHN [1], even when directional information is entirely disregarded. This is evidenced by DSHN achieving higher mean accuracy in 5 out of 7 datasets. These results clearly highlight the importance of the formal properties associated with our generalized Laplacian operator (Section 3.3). This structural rigor is the key to the robustness seen across all spectral-based approaches, including established methods like GCN, MagNet, and graph-based SNNs.
> >
> > > Could you be more specific regarding the novelty of the objects you use in your definitions? For example, it is not clear to me whether the object $\mathcal{S}^{(q)}$ represents a novel contribution or not.
> >
> > A central contribution of our work is the introduction of a fully tunable complex-valued coefficient associated with each node-hyperedge incidence, encoded through the directional matrix $\mathcal{S}^{(q)}$ which has deep connections with the Magnetic Laplacian. Through such definition, as we show in the paper, we can introduce directional information into the Directed Sheaf Hypergraph Laplacian, recovering as special cases many previous operators from the graph- and hypergraph-learning literature such as MagNet and GeDi-HNN.
> > Moreover, our framework equips the hypergraph with a Cellular Sheaf. By learning optimal restriction maps, the model can regulate information flow across hyperedges, thereby mitigating oversmoothing and improving performance under heterophilic settings. Each node can thus contribute differently to each group it participates in, enabling richer expressivity than prior directed and undirected hypergraph neural networks.
> > Finally, as we discussed in our paper, and addressed in the paragraph above, our Directed Sheaf Hypergraph Laplacian is also (to the best of our knowledge) the first sheaf-based convolutional operator enjoying all the desired spectral properties applicable to both directed and undirected hypergraphs.

---

> > > ### Author Response · Authors · 2025-11-20
> > > **Response to Reviewer YUAy - Part 3**
> > >
> > > >In the experimental section, there is no introduction or citation of some of the most relevant baselines.
> > >
> > > In the original submission we had included descriptions of these baselines in the Appendix due to space constraints. With the additional page now available, we have added citations to relevant baselines directly in Section 5. For a more detailed discussion of these methods and their characteristics, we still refer readers to the Appendix.
> > >
> > > >Too little space is dedicated to properly introducing the background and previous methods.
> > >
> > > With the original 9-page limit, we had to carefully balance technical depth with accessibility. With the additional page now available, we have expanded Sections 2 and 3 to provide more background material and a more detailed discussion of SHN.  Specifically, Section 2 now includes a more comprehensive introduction to the foundational concepts of cellular sheaves and sheaf neural networks, along with a clearer explanation of our notation. In Section 3, we have added a detailed discussion comparing our approach to SHN, explicitly addressing the differences between the two methods and clarifying why certain properties hold for DSHN but not for SHN.
> > >
> > > >I recommend to define more thoroughly the notation used in Section 2.
> > >
> > > We have carefully revised the manuscript to simplify the notation while maintaining conceptual clarity. Following the reviewer's suggestion, we now use different fonts to distinguish between matrices and vectors, which should make the mathematical expressions easier to follow throughout the paper. We have also added a dedicated paragraph at the end of Section 2 that clearly explains the notational conventions used in the manuscript. Finally, the inconsistency between $\delta_e$ and $\delta(e)$ was solved.
> > >
> > > **Bibliography:**
> > >
> > > [1] Duta et al., Sheaf hypergraph networks, 2023.

---

> ### Author Response · Authors · 2025-11-27
>
> Dear Reviewer **YUAy**,
> Thank you once again for your thoughtful and constructive evaluation of our work. Your questions provided opportunities to clarify and further strengthen the presentation and positioning of the paper. We have made the necessary revisions accordingly.
>
> We believe we have addressed all your concerns in our responses, which were submitted a week ago. However, we have not yet received any feedback. We would be grateful for any further comments you might have, as your perspective would help us ensure that everything is resolved.
>
> The Authors

---

### Official Review · Reviewer_SDBh · 2025-10-30

**Soundness:** 3
**Presentation:** 2
**Contribution:** 3
**Rating:** 6
**Confidence:** 2

**Summary:**

The paper introduces the concept of directed hypergraph cellular sheaves, extending sheaf neural networks to directed hypergraphs. The paper proposes a directed sheaf hypergraph Laplacian, a principled spectral operator. Finally, the paper introduces DSHNN, combining sheaf theory with directional information, enabling strong performance on homophilous/heterophilous graphs.

**Strengths:**

### Strengths

- **Interesting technical problem.** Extending sheaf neural networks to directed hypergraphs is an interesting technical problem.
- **Principled operator design and theoretical foundations.** The work provides a directed sheaf–hypergraph Laplacian with principled spectral properties and a unifying view that recovers prior (graph/hypergraph/magnetic) operators as special cases. The theoretical results give a well-posed basis for diffusion/convolution with per-incidence maps and directional phases.
- **Empirical competitiveness within the sheaf/hypergraph family.** The proposed models outperform or match prior sheaf/hypergraph baselines on a range of benchmarks.

**Weaknesses:**

### Weaknesses

- **High-level intuition is hard to understand.** For a reader unfamiliar with sheaf neural networks, I struggled to understand some of the motivation and would appreciate if the paper expanded on the intuition for the approach. One example of where I think expanding on intuition would help is why sheaves mitigate oversmoothing, oversquashing, and heterophily. This would help the reader appreciate the goal to extend them to directed hypergraph case. Another is if the paper could expand at a high level why the theoretical results are important. Why is it necessary that DSHN satisfies all the proven properties? Though each of these results are sound and clearly presented, I struggled to understand why each of these are desirable to have.
- **Empirical setup and baselines.** I am not convinced by the decision to process datasets such as Cora, Chameleon, and Squirrel as directed hypergraphs. What does this star-expansion gain over modeling the dataset as an undirected graph (how the datasets are usually processed)? Additionally, if the authors are aware of or could evaluate on datasets with native directed-hypergraph representations (where one-to-many interactions are intrinsic), this would strengthen the empirical evaluation. As it stands now, the datasets evaluated on are usually processed as undirected graphs and I'm unsure what the issue is with these standard ways of processing them. Finally, I would appreciate if the authors could justify when to use DSHN over standard MPNNs and MPNNs designed for heterophily. Currently, there is no discussion of or empirical comparison against against strong heterophilous GNNs (e.g., H2GCN, GCNII). I think these should be discussed and ideally evaluated against since much effort has been dedicated to these types of GNNs for homophilous and heterophilous datasets.

**Questions:**

My questions are integrated with the weaknesses section.

---

> ### Author Response · Authors · 2025-11-20
> **Response to Reviewer SDBh - Part 1**
>
> >For a reader unfamiliar with sheaf neural networks, I struggled to understand some of the motivation and would appreciate if the paper expanded on the intuition for the approach. One example of where I think expanding on intuition would help is why sheaves mitigate oversmoothing, oversquashing, and heterophily. This would help the reader appreciate the goal to extend them to directed hypergraph case.
>
> We have added a dedicated paragraph in Section 2 to clarify why Sheaves mitigate oversmoothing and offer greater expressivity than traditional Graph and Hypergraph Neural Networks in heterophilic settings. We additionally provide in the following paragraph a discussion on such properties at an intuitive level.
>
> When a Cellular Sheaf is attached to a graph or hypergraph, higher-dimensional vector spaces are assigned to both nodes and hyperedges. As a consequence, a node’s representation is no longer forced to resemble that of its neighbors, a limitation that typically arises in standard graph and hypergraph neural networks biased toward homophily (e.g., GCN[1], HGNN[2]). With a Cellular Sheaf, each node can maintain its own representation, while restriction maps, linear transformations between node and (hyper-)edge vector spaces,  govern how information is transferred across the structure.
> As highlighted in Section 2, this perspective can be understood through the lens of opinion dynamics [3]: each node holds its own “opinion”, and restriction maps determine how this private information is communicated to adjacent nodes, potentially in different ways depending on the hyperedge (or group) involved.
> Through this mechanism, sheaf-based models naturally mitigate oversmoothing and are particularly effective in heterophilic settings. In contrast to traditional message-passing methods, where adjacent nodes are forced to become similar regardless of whether they actually share similar features (or labels), restriction maps explicitly mediate how information is transferred between adjacent nodes. By learning a distinct linear transformation for each node-edge incidence, the model can "translate" a node’s features differently to each edge it participates in. This also prevents nodes' representation from collapsing, as the number of layers of the architecture increases, preventing oversmoothing.
>
> > Another is if the paper could expand at a high level why the theoretical results are important. Why is it necessary that DSHN satisfies all the proven properties? Though each of these results are sound and clearly presented, I struggled to understand why each of these are desirable to have.
>
> We have added a dedicated explanation in Section 3.3. All changes are highlighted in blue in the updated PDF to facilitate your review.
>
> The spectral properties we prove are not merely formal requirements but they ensure that the Fourier transform is well-defined and that polynomial filters of $\mathbf{L}^{\vec{\mathcal{F}}}_N$ implement localized, stable convolutions, in direct analogy with classical spectral-based approaches [1, 7]. Failure to preserve these properties can result in fundamental issues, and more specifically a complete breakdown of the connection between message passing, the Fourier transform of a (hyper-)graph and its connections with signal theory. Let us note that all these properties are enjoyed by the classical graph Laplacian $\mathbf{L = D - A}$.
>
> Moreover, our Generalization Properties section shows that the Directed Sheaf Hypergraph Laplacian recovers, as special cases, a wide range of existing operators from the graph- and hypergraph-learning literature. In order to ensure that our operator can unify and theoretically subsume many prior methods, such as HGNN [2], GeDi-HNN [4], SNN [5,6] and MagNet [8], we must prove that our object has all the required spectral properties that are also shared with prior convolutional operators. See also the additional experiments we conducted for reviewer **YUAy** (Part 2).

---

> > ### Author Response · Authors · 2025-11-20
> > **Response to Reviewer SDBh - Part 2**
> >
> > >What does this star-expansion gain over modeling the dataset as an undirected graph (how the datasets are usually processed)?
> >
> > Although datasets such as Cora, Chameleon and Squirrel are often treated as undirected in different works [1, 11], their original structure is in fact directed, since citation links naturally encode one-way relationships. Our star-expansion preserves this underlying directionality pattern of the data, making the original directed graph data expressed as a directed hypergraph. It rewrites each node’s outgoing neighbors as a directed hyperedge, which keeps the directional information of the original data intact while expressing it in a form suitable for hypergraph-based models. This representation is also common in prior hypergraph learning works [2,4], and using it allows us to compare our method to these baselines under consistent preprocessing assumptions. Moreover, it is a widely adopted strategy in the hypergraph-learning literature to use original graph data to model hypergraphs [9, 10].
> >
> > Finally notice that, for our method, by "compressing" the representation of the out-neighborhood of a node into a single directed hyperedge, the number of restriction maps to be learned becomes smaller. In fact, let's consider an original directed graph with 4 nodes {A,B,C,D} in total and 3 nodes connected to the 4th one through 3 directed edges that go from D to A, B and C separately. In this scenario the restriction maps to be employed become two for incidence relation: every edge is incident with two nodes, hence a total of 6 restriction maps. While, on the other hand, by rewriting it as a single directed hyperedge, the number of maps to be learned becomes just 4, one for each node-hyperedge incident relation.
> >
> > >Finally, I would appreciate if the authors could justify when to use DSHN over standard MPNNs and MPNNs designed for heterophily. Currently, there is no discussion of or empirical comparison against against strong heterophilous GNNs (e.g., H2GCN, GCNII). I think these should be discussed and ideally evaluated against since much effort has been dedicated to these types of GNNs for homophilous and heterophilous datasets.
> >
> > Regarding the comparison with heterophilous MPNNs, our method is designed for both directed and undirected hypergraphs, which generalize graphs by allowing each hyperedge to connect an arbitrary number of nodes rather than just two (which are not feasible for traditional GNNs).
> >
> > We want to emphasize, however, that a direct empirical comparison with heterophilous GNNs is *_not straightforward_*: even on the same dataset, the underlying topology differs significantly (hypergraphs vs. pairwise graphs).
> > Although our method can also be, in principle, employed for the special case of directed graphs (2-uniform directed hypergraphs), as highlighted in the Generalization Properties section, we did not explore this direction as it falls outside the scope of our current work.
> > Nevertheless, for reference, we provide some comparisons against strong heterophilous graph neural networks (executed on the graph version of the datasets) and our proposed DSHN (executed on the directed hypergraph representation).
> >
> > Table 1: Node classification accuracy (%) comparison between heterophilous GNNs (on graph topology) and our DSHN (on directed hypergraph topology)
> > | Method                     | Roman              | Squirrel           | Telegram           | Cora               | Citeseer            |
> > |----------------------------|--------------------|--------------------|--------------------|--------------------|----------------------|
> > | H2GCN (Graph)              | 60.14 ± 0.55       | 37.62 ± 1.98       | 88.11 ± 3.92       | 87.73 ± 1.24       | 77.05 ± 1.60         |
> > | GCNII (Graph)              | 83.62 ± 0.49       | 42.11 ± 2.18       | **89.15 ± 3.88**   | **88.45 ± 1.28**   | 77.28 ± 1.52         |
> > | FAGCN (Graph)              | 74.69 ± 0.75       | 40.97 ± 2.22       | 80.89 ± 7.72       | 88.25 ± 1.21       | 77.17 ± 1.79         |
> > | DSHN (Directed-hypergraph) | **89.24 ± 0.57**   | **44.09 ± 2.36**   | 88.65 ± 5.54       | 88.02 ± 1.11       | **77.45 ± 0.74**     |
> >
> >
> > In this comparison, where DSHN is tasked with a fundamentally different data structure, our method manages to outperform existing heterophilous GNNs on the original graph representation in 3 out of 5 cases (Roman, Squirrel, Citeseer). Moreover, in the remaining two cases (Telegram and Cora), its performance is in line with the best baseline models.

---

> > > ### Author Response · Authors · 2025-11-20
> > > **Response to Reviewer SDBh - Part 3**
> > >
> > > **Bibliography**
> > >
> > > [1] Thomas N. Kipf, Max Welling, Semi-Supervised Classification with Graph Convolutional Networks, ICLR 2017.
> > >
> > > [2] Feng, Y., You, H., Zhang, Z., Ji, R., & Gao, Y. (2019). Hypergraph Neural Networks. Proceedings of the AAAI Conference on Artificial Intelligence, 33(01), 3558-3565.
> > >
> > > [3] Hansen, J., & Ghrist, R. (2021). Opinion dynamics on discourse sheaves. SIAM Journal on Applied Mathematics, 81(5), 2033–2060.
> > >
> > > [4] Fiorini, S., Coniglio, S., Ciavotta, M., & Del Bue, A. (2024). Let There be Direction in Hypergraph Neural Networks. Transactions on Machine Learning Research.
> > >
> > > [5] Bodnar, C., Di Giovanni, F., Chamberlain, B. P., Lio, P., & Bronstein, M. M. (2022). Neural sheaf diffusion: A topological perspective on heterophily and oversmoothing in GNNs. In A. H. Oh, A. Agarwal, D. Belgrave, & K. Cho (Eds.), Advances in Neural Information Processing Systems.
> > >
> > > [6] Jakob Hansen and Thomas Gebhart. Sheaf neural networks, 2020.
> > >
> > > [7] Defferrard, M., Bresson, X., & Vandergheynst, P. (2016). Convolutional neural networks on graphs with fast localized spectral filtering. In Advances in Neural Information Processing Systems.
> > >
> > > [8] Zhang, X., He, Y., Brugnone, N., Perlmutter, M., & Hirn, M. (2021). MagNet: A neural network for directed graphs. In M. Ranzato, A. Beygelzimer, Y. Dauphin, P. S. Liang, & J. Wortman Vaughan (Eds.), Advances in Neural Information Processing Systems (Vol. 34, pp. 27003–27015).
> > >
> > > [9] Yadati, N., Nimishakavi, M., Yadav, P., Nitin, V., Louis, A., & Talukdar, P. (2019). HyperGCN: A new method of training graph convolutional networks on hypergraphs. In Proceedings of the 33rd International Conference on Neural Information Processing Systems (Art. 135, 12 pages). Curran Associates Inc.
> > >
> > > [10] Li, M., Gu, Y., Wang, Y., Fang, Y., Bai, L., Zhuang, X., & Liò, P. (2025). When Hypergraph Meets Heterophily: New Benchmark Datasets and Baseline. Proceedings of the AAAI Conference on Artificial Intelligence, 39(17), 18377-18384.
> > >
> > > [11] Bodnar, C., Di Giovanni, F., Chamberlain, B. P., Lio, P., & Bronstein, M. M. (2022). Neural sheaf diffusion: A topological perspective on heterophily and oversmoothing in GNNs. In A. H. Oh, A. Agarwal, D. Belgrave, & K. Cho (Eds.), Advances in Neural Information Processing Systems.

---

> ### Author Response · Authors · 2025-11-27
>
> Dear Reviewer **SDBh**,
> Thank you once again for your thoughtful and constructive evaluation of our work. Your questions provided opportunities to clarify and further strengthen the presentation and positioning of the paper. We have made the necessary revisions accordingly.
>
> We believe we have addressed all your concerns in our responses, which were submitted a week ago. However, we have not yet received any feedback. We would be grateful for any further comments you might have, as your perspective would help us ensure that everything is resolved.
>
> The Authors

---

### Official Review · Reviewer_rWec · 2025-10-31

**Soundness:** 3
**Presentation:** 2
**Contribution:** 2
**Rating:** 4
**Confidence:** 3

**Summary:**

This paper extended Sheaf Neural Networks to work with directed hypergraphs. The authors define Cellular Sheaves for directed hypergraphs and then the Laplacian matrix associated with it. The authors show that the Laplacian matrix associated with directed sheaf hypergraphs has some favourable properties such as being diagonalizable with real eigenvalues, positive semidefinite, and has bounded spectrum. The authors also show that the new Laplacian matrix generalizes several previously considered special cases, including the classical graph Laplacian, the Magnetic Laplacian for directed graphs, and some existing hypergraph Laplacians. By parameterizing the heat diffusion induced by the new Laplacian operator, the authors obtain a neural network architecture which they call Directional Sheaf Hypergraph Network (DSHN). The authors show that DSHN outperforms several baselines over both real-world and synthetic datasets.

**Strengths:**

The Laplacian operator for directed sheaf hypergraphs is new and generalizes several previously considered special cases (but the extension is straightforward).

The experiments are reasonably comprehensive and well presented. Compared with existing baselines, overall there is a notable improvement in accuracy. Figure 2 clearly demonstrates the benefits of taking directionality into account and having a hyperparameter q which adjusts the level of directionality to be used.

Overall, the paper presents a simple yet seemingly effective idea to solve node classification tasks on directed hypergraphs.

**Weaknesses:**

The presentation and clarity can be improved. In particular, for readers who are not very familiar with cellular sheaves and Sheaf Neural Networks, the notations are a bit heavy and difficult to parse. The authors should try not to overload the text with unnecessary and complex notations. Maybe using different fonts for matrices/vector spaces/functions/maps could help. The choice of notations should be clearly stated at the beginning.

(This is not necessarily a weakness of this paper, but my rating is definitely affected by this) For several years I have been reviewing papers that propose new GNNs to solve node classifications tasks on standard benchmarks, and over time I start to get bored at reading papers that aim to improve node classification accuracy on standard benchmarks only. Indeed, with new methods like the one introduced in this paper, you can improve the performance on benchmarks. But really, how many of these new methods are practically relevant at all (i.e. being implemented in practice to solve actual tasks)? The benchmarks are so poor that many of the node features are still bag-of-words one-hot encodings. Even if a new method performs better on these benchmarks, I am not sure how well this will generalize to practical settings, where you can at least use a language model to generate much better features from raw texts. Because of this, I think the practical relevancy of this work is relatively low. Also see the recent position paper [1]. This work would have been great if the authors also develop some new benchmarks that are more aligned with the current ML practice (e.g. use better features).

Refs:\
[1] M. Bechler-Speicher et al. Position: Graph Learning Will Lose Relevance Due To Poor Benchmarks. ICML 2025

**Questions:**

None

---

> ### Author Response · Authors · 2025-11-20
> **Response to Reviewer rWec - Part 1**
>
> > For several years I have been reviewing papers that propose new GNNs to solve node classifications tasks on standard benchmarks, and over time I start to get bored at reading papers that aim to improve node classification accuracy on standard benchmarks only. Indeed, with new methods like the one introduced in this paper, you can improve the performance on benchmarks. [...] Even if a new method performs better on these benchmarks, I am not sure how well this will generalize to practical settings, where you can at least use a language model to generate much better features from raw texts. Because of this, I think the practical relevancy of this work is relatively low.
>
> We believe that comparing against existing benchmarks remains essential for demonstrating that a new method works and for enabling fair comparison with prior work. These benchmarks provide a common ground for evaluating different approaches. Without this comparison, it would be difficult to assess whether our method offers any advantage over existing techniques. In this sense, we believe that the applicability of our method is on-par with previous methods from the hypergraph-learning literature since, as highlighted in our work, it can handle both directed and undirected hypergraphs.
>
> To directly address the reviewer's concern regarding practical relevance and the generalization capability of our approach, we have performed an additional study on two datasets involving natively directed hypergraphs for molecular reactions. This new evaluation is framed as a hyperedge classification task (a task not considered initially in our work), where the goal is to predict the class of the reaction represented by the hyperedge. We model the reaction data as a directed hypergraph, inspired by the methodology in [1]. In this model, each directed hyperedge represents a reaction, structured into a set of reactants (the hyperedge's tail set) and a set of products (the hyperedge's head set). These datasets are derived from subsets of the ORD and USTPO datasets, which are popular open-source resources for retrosynthesis problems.
>
> The results reported in Table 1 validate the practical applicability of our method to real-world hyperedge classification tasks, such as molecular reaction prediction. Performance is measured using the F1-score, accounting for the high imbalance in the datasets. Comprehensive details on the datasets and training protocol are provided in the Supplementary Material.
>
> Table 1: Hyperedge classification F1-score results for prediction of molecular reaction types on two hypergraph datasets
>
> | Method              | Dataset-1              | Dataset-2              |
> |---------------------|-----------------------|-----------------------|
> | HGNN                | 69.38 $\pm$ 0.48          | 81.40 $\pm$ 2.68          |
> | HNHN                | 32.27 $\pm$ 1.30          | 45.69 $\pm$ 7.48          |
> | UniGCNII            | 72.00 $\pm$ 0.59          | 85.61 $\pm$ 2.63          |
> | LEGCN               | OOM                    | 84.75 $\pm$ 2.68          |
> | HyperND             | 44.16 $\pm$ 1.27          | 82.86 $\pm$ 3.17          |
> | AllDeepSets         | 79.17 $\pm$ 0.53          | 85.78 $\pm$ 3.01          |
> | AllSetTransformer   | 79.24 $\pm$ 1.08          | $\underline{87.89 \pm 2.87}$   |
> | ED-HNN              | 66.37 $\pm$ 2.62          | 87.05 $\pm$ 1.96          |
> | SheafHyperGNN       | 57.99 $\pm$ 2.75          | 80.25 $\pm$ 2.20          |
> | PhenomNN            | 47.71 $\pm$ 2.90          | 86.27 $\pm$ 2.40          |
> | GeDi-HNN            | $\underline{80.72 \pm 0.78}$   | 85.64 $\pm$ 2.42          |
> | DHGNN               | OOM                    | 85.93 $\pm$ 3.49          |
> | DSHN                | OOM     | **89.09 $\pm$ 3.08**      |
> | DSHNLight           | **82.32 $\pm$ 0.56**      | 86.52 $\pm$ 2.68          |
>
>
> We will include the use of LM-generated features as a possible direction for future work.

---

> > ### Author Response · Authors · 2025-11-20
> > **Response to Reviewer rWec - Part 2**
> >
> > >The presentation and clarity can be improved. In particular, for readers who are not very familiar with cellular sheaves and Sheaf Neural Networks, the notations are a bit heavy and difficult to parse. The authors should try not to overload the text with unnecessary and complex notations. Maybe using different fonts for matrices/vector spaces/functions/maps could help. The choice of notations should be clearly stated at the beginning.
> >
> > We have carefully revised the manuscript to simplify the notation while maintaining conceptual clarity. However, we would like also to highlight that prior work on Sheaf Neural Networks has similar notation [2], which we agree can be a bit heavy (but for continuity with prior work, we must employ). Following the reviewer's suggestion, we now use different fonts to distinguish between matrices and vectors, which should make the mathematical expressions easier to follow throughout the paper. We have also added a dedicated paragraph at the end of Section 2 that clearly explains the notation used in the manuscript. All changes are highlighted in blue in the updated PDF to facilitate your review.
> >
> > **Bibliography:**
> >
> > [1] Restrepo, G. Spaces of mathematical chemistry. Theory Biosci. 143, 237–251 (2024). https://doi.org/10.1007/s12064-024-00425-4
> >
> > [2] Bodnar, C., Di Giovanni, F., Chamberlain, B. P., Lio, P., & Bronstein, M. M. (2022). Neural sheaf diffusion: A topological perspective on heterophily and oversmoothing in GNNs. In A. H. Oh, A. Agarwal, D. Belgrave, & K. Cho (Eds.), Advances in Neural Information Processing Systems.

---

> > > ### Author Response · Authors · 2025-11-27
> > >
> > > Dear Reviewer **rWec**,
> > > Thank you once again for your thoughtful and constructive evaluation of our work. Your questions provided opportunities to clarify and further strengthen the presentation and positioning of the paper. We have made the necessary revisions accordingly.
> > >
> > > We believe we have addressed all your concerns in our responses, which were submitted a week ago. However, we have not yet received any feedback. We would be grateful for any further comments you might have, as your perspective would help us ensure that everything is resolved.
> > >
> > > The Authors

---

### Author Response · Authors · 2025-12-02
**Authors' Summary for AC**

Dear Area Chair,

Thank you for managing the review process. We are grateful to the reviewers for their overall positive evaluation of our work. The reviewers highlighted several key **strengths** of the paper:

* **Theoretical Rigor:** They noted that the proposed **Directed Sheaf Hypergraph Laplacian** provides a **principled extension** of prior operators, confirming that the proofs are clear and well explained.
* **Motivation and Problem Framing:** Reviewers found the overall motivation, task, and goals to be clear, and found the benefits of **incorporating directionality** and tuning the associated hyperparameter to be well illustrated.
* **Experimental Results:** The experiments were considered **thorough and clearly presented**, consistently demonstrating improvements over existing baselines from the directed and undirected hypergraph-learning literature.

In response to the reviewers' comments, we provided detailed answers and incorporated new experimental results to assess and solidify the strength of our work. Specifically, we have focused on:

* **Practical Relevance and New Datasets:** We conducted additional experiments on two new directed and real-world datasets focused on molecular reaction type prediction (a hyperedge classification task) to directly address concerns regarding real-world applicability beyond the original benchmarks considered for our work. We observed improvements over existing methods, validating the practical utility of our approach. Further details are provided in the Part 1 response to **rWec** and in the Supplementary Material.

* **Presentation and Accessibility Improvement:** We improved the clarity of the presentation by explaining the notation employed throughout the work
(new paragraph in Section 2). Additionally, following the suggestion of reviewer **YUAy**, we now distinguish different objects using separate font types to improve readability.

* **Novelty Clarification:** Reviewer **YUAy** requested more specificity regarding the novelty of the $\mathcal{S}^{(q)}$ object. $\mathcal{S}^{(q)}$ is a central contribution of our work, as it enables the introduction of a fully tunable complex-valued coefficient associated with each node–hyperedge incidence relation. Through this definition, we can incorporate directional information into the Directed Sheaf Hypergraph Laplacian operator, recovering as special cases many previous operators from the graph- and hypergraph-learning literature. We provided a detailed summary in our response and added clarification in Section 3, articulating the novelty of $\mathcal{S}^{(q)}$ and its connections to prior work.

* **Necessity of Properties and Comparison with SHN:** We addressed questions (from **SDBh** and **YUAy**) regarding the necessity of our Laplacian’s proven properties (e.g., positive semidefiniteness) by explaining in our responses why they are important and how they distinguish our operator from the one proposed in Sheaf Hypergraph Networks. To assess their practical impact, we conducted additional experiments comparing DSHN and SHN in the absence of directional information, showing that DSHN is significantly more stable and expressive due to its associated properties, as reported in the response to **YUAy** Part 2. We added further details on the properties and their significance in a paragraph in Section 3, while Appendix E provides an example where the SHN's operator fails to maintain some of these spectral guarantees.



* **Comparison with GNNs/MPNNs:** Reviewer **SDBh** questioned the necessity of our star-expansion procedure for converting directed graphs into hypergraphs and suggested comparisons with strong heterophilous GNNs. We addressed these points: (1) the star-expansion procedure aligns with prior hypergraph-learning work, and (2) additional experiments show that our method outperforms or matches strong heterophilous GNNs/MPNNs despite differing underlying data structures. Details are in the table provided in **SDBh** Part 2.


* **Computational Complexity:** Some reviewers raised questions about the computational complexity of our method. Some experiments in this regard were included in the original Supplementary Material, and we now report the complexity analysis also in the main paper. Figure 4 in the Appendix shows that, for models with similar parameter budgets, our method consistently outperforms baselines, indicating that improvements are not due to model size alone. Moreover, following reviewers' suggestions, we provide a FLOPs comparison with other sheaf and non-sheaf baselines in Table 7 of the Supplementary Material.

Overall, we believe the discussion period allowed us to effectively address the reviewers’ comments through additional experiments, clarifications, and revisions and we hope that these updates provide the Area Chair with a clearer view of the contribution our work.

Best regards,

The Authors

---

### Meta-Review · Area_Chair_TAe1 · 2026-01-06

**Summary:**

**Summary:**
This paper introduces Directional Sheaf Hypergraph Networks which extends sheaf neural networks to directed hyper graphs. The authors do this through the use of a complex valued restriction map, and because this gives a generalization introduce a unifying framework that encompasses many existing laplacian formulations. The method yields significant improvements across 7 different datasets.

**Rationale:**
The paper makes solid theoretical contributions by exploring an extension of sheaf theory to directed hyper graphs. The author addressed any of the concerns through their rebuttal but concerns about practical relevance remain. This concern is evergreen in the graph literature, however. In all, the reviews likely were a 5/6/6 which indicates a borderline accept, and in my opinion the paper was interesting.

**Reviewer Concerns:**

rwec:
- Practical relevance: The authors addressed this concern as best as they could by adding additional datasets, but this is a problem for the graph community at large and the paper shouldn't be penalized for it
- Boredom with benchmark papers: inappropriate comment by the reviewer
- Heavy notation: addressed in spots, but I also don't agree with the reviewer

sdbh:
- Comparison with heterophilic GNNs: Added, and also clarified as to why this was not included in the first place (subtly different task)
- Complex valued approach: Expanded the intuition as to why this is needed.
- Star expansion: Added justification

yuay:
- Computational overhead: Added a table in the appendix
- Lack of natively directed hyper graph datasets: Authors added reaction networks (cool find!)
- Novelty of complex laplacian: addressed
- Missing SHN comparison: addressed, even when q=0

**Reviewer Scores:**

- rwec, 4->5
- sdbh, 6->6, the critiques were minor so the scores likely wouldn't have changed
- yuay, 6->6, the critiques were minor so the scores likely wouldn't have changed

---

### Decision · Program_Chairs · 2026-01-26

Accept (Poster)